# D4Explainer: In-Distribution GNN Explanations via Discrete Denoising Diffusion

**Jialin Chen**
Yale University
jialin.chen@yale.edu

**Shirley Wu**
Stanford University
shirwu@cs.stanford.edu

**Abhijit Gupta**
Yale University
abhijit.gupta@yale.edu

**Rex Ying**
Yale University
rex.ying@yale.edu

## Abstract

The widespread deployment of Graph Neural Networks (GNNs) sparks significant interest in their explainability, which plays a vital role in model auditing and ensuring trustworthy graph learning. The objective of GNN explainability is to discern the underlying graph structures that have the most significant impact on model predictions. Ensuring that explanations generated are reliable necessitates consideration of the in-distribution property, particularly due to the vulnerability of GNNs to out-of-distribution data. Unfortunately, prevailing explainability methods tend to constrain the generated explanations to the structure of the original graph, thereby downplaying the significance of the in-distribution property and resulting in explanations that lack reliability. To address these challenges, we propose D4Explainer, a novel approach that provides in-distribution GNN explanations for both counterfactual and model-level explanation scenarios. The proposed D4Explainer incorporates generative graph distribution learning into the optimization objective, which accomplishes two goals: 1) generate a collection of diverse counterfactual graphs that conform to the in-distribution property for a given instance, and 2) identify the most discriminative graph patterns that contribute to a specific class prediction, thus serving as model-level explanations. It is worth mentioning that D4Explainer is the first unified framework that combines both counterfactual and model-level explanations. Empirical evaluations conducted on synthetic and real-world datasets provide compelling evidence of the state-of-the-art performance achieved by D4Explainer in terms of explanation accuracy, faithfulness, diversity, and robustness. [1]

## 1 Introduction

Graph neural networks (GNNs) have rapidly gained popularity recently due to their ability to model relational data [1, 2]. However, when it comes to critical decision-making and high-stake applications, such as healthcare, finance, and autonomous systems, the explainability of GNNs is fundamental for humans to understand the model's decision-making logic and build trust in the deployment of GNNs in real-world scenarios [3, 4, 5].

**Counterfactual and model-level explanations**. Existing methods mainly focus on factual and instance-level explanations [6, 7, 8, 9, 10, 11], while the significance of counterfactual and model-level explanations are equally noteworthy, yet under-explored. Counterfactual explanation considers

---

[1]The code is available at https://github.com/Graph-and-Geometric-Learning/D4Explainer

37th Conference on Neural Information Processing Systems (NeurIPS 2023).

"what-if" scenarios of model predictions, addressing the question of how slight adjustments to the input graph can lead to different model predictions [12, 13, 14]. Model-level explanation, on the other hand, aims to generate the most discriminative graph pattern for a target class, thus shedding light on the overall decision-making behavior and internal functioning of the model [15, 16]. Counterfactual and model-level explanations present a distinct challenge concerning the distribution constraint imposed on generated explanations. An explanation that is faithful and reliable should adhere to the distribution of the underlying dataset. This becomes particularly crucial in real-world scenarios where domain-specific rules exist, such as in drug design and molecule generation. In such cases, explanations should conform to the true distribution of the dataset [16, 17, 18].

However, the existing methods typically extract explanatory subgraphs from the input graph, ignoring additional possible edges. This prevailing paradigm heavily relies on the out-of-distribution (OOD) effect to influence the model's prediction. To illustrate this point, in Figure 1, we show the t-SNE projection of the Tree-Cycle dataset, where graphs are labeled as *Tree* or *Cycle* based on whether they present the corresponding structures. Specifically, CF-GNNExplainer [13] generates counterfactual explanations for a node with *Tree* label by removing its neighbor edges. While the explanation doesn't maintain any discriminative information on the Cycle class, it could still be predicted as *Cycle* with high probability due to the OOD effect, making the explanation unreliable.

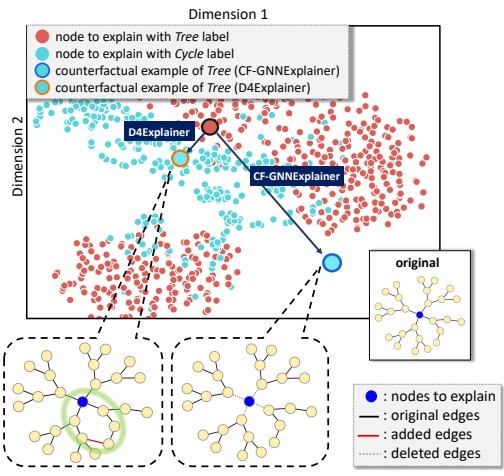

Figure 1: t-SNE Projection of Tree-Cycle dataset, where *Cycle* is a counterfactual motif for *Tree*.

On the other hand, generating in-distribution graphs is challenging, due to the difficulty of encoding complex graph distributions, *e.g.,* the distribution of node degrees, cycle counts and edge homogeneity. Recently, graph diffusion models have shown to be a powerful technique to encode such complex distribution on graphs [19, 20], which trains a powerful denoising model that progressively removes noise from a random noise graph and then tractably recovers in-distribution samples.

**Proposed work**. Inspired by the success of graph diffusion models, we propose a novel GNN explainability approach, **D4Explainer**, in-**D**istribution GNN explanations via **D**iscrete **D**enoising **D**iffusion. Through a forward diffusion process that progressively introduces random noise, we enable D4Explainer to optimize for alternative and diverse explanations based on multiple noisy versions of the given graph. A powerful denoising model is trained to remove noise and eliminate redundant edges that are irrelevant to the target property, thereby ensuring the model's robustness. By employing a carefully designed loss function that incorporates both the preservation of the counterfactual property and generative graph distribution learning, D4Explainer is capable of generating in-distribution counterfactual explanations. As highlighted in green in the bottom left of Figure 1, D4Explainer adds essential edges that complete the truly counterfactual motif, *i.e., Cycle*. With a slight modification to the loss function, D4Explainer can also perform model-level explanations for a specific target class.

Empirical experiments on eight synthetic and real-world datasets show that D4Explainer achieves state-of-the-art performance in both counterfactual and model-level explanations, with a strong counterfactual accuracy (>80% ) when only 5% of the edges are modified. Maximum mean discrepancy (MMD) metrics show that the distribution of explanations generated by D4Explainer is the closest to the original distribution of the dataset, compared with all baselines. D4Explainer obtains the highest Top-$K$ accuracy in the robustness evaluation, which further illustrates that D4Explainer is capable of generating consistent explanations with the presence of noise.

**Our contributions** are in three-folds: (1) A novel approach to generate in-distribution, diverse and robust explanations is proposed, which leverages the denoising diffusion model to capture the underlying distributions of explanation graphs; (2) D4Explainer explores counterfactual explanations in a larger search space by allowing adding edges, which provides high-level understandings of how edge addition helps to create truly counterfactual motifs; (3) D4Explainer represents the first

framework that unifies counterfactual and model-level explanations, providing faithful explanations for both settings.

## 2    Related Work

**Explainability of GNNs**    Compared with the explainability methods in image domain [21, 22, 23, 24, 25, 26, 27], explainability in GNNs [3] remains a challenging problem due to the discrete structure of graphs. Here, we focus on post-hoc and model-agnostic explanations. **Non-parameterized methods** rely on gradient-like signals [28, 29], relevant walks [30, 31], perturbation [9, 32, 33, 34] to identify important node/edge features or graph structures as explanations, without learnable parameters. **Score-based explainability** [6, 8, 7, 35] formulate a trainable model to obtain the importance scores on node features or edge as the explanations by maximizing the mutual information between the explanatory subgraph and the target prediction. **Counterfactual explanation methods** find minimal perturbation to the graph instance such that the prediction changes. However, most existing methods [13, 36] only consider edge deletion on the original graph without any distribution constraints, thus easily creating out-of-distribution samples and overfitting the noise over each individual instance. CLEAR [37] is the only explainer that also considers adding edges in generating counterfactual explanations. However, the intrinsic effect of edge addition to counterfactual properties is under-explored by CLEAR. **Generation-based explanations** is a recently popular trend that trains graph generators to generate GNN explanations. Existing works train policy networks for the sequential graph generation process based on the reinforcement learning approach [38, 10, 15] or explicitly parameterize the distribution of model-level explanations [16]. The differences of our method are (1) we prevent explicit modeling and sequential decision-making learning but incorporate the generative graph distribution learning implicitly into the training procedure and (2) the more stable and robust generative backbone *i.e.,* diffusion model ensures better properties of the generated explanations, *e.g.,* diversity and robustness.

**Graph Diffusion Models**    Denoising diffusion probabilistic models [39, 40, 41] are shown to be powerful for a wide range of generative tasks, including images [42], language [43], and discrete graph domain [19, 20, 44]. Recent work [20] proposes to use discrete noise for the forward Markov process without relying on continuous Gaussian perturbations. Another related work [19] formulates the diffusion process on the categorical node and edge attributes and successfully generates real and in-distribution graphs. Recently, the score-based model [45] and stochastic differential equations formulation have been applied to the field of graph generation [46, 47]. These related works highlight the effectiveness of diffusion models for graph denoising and generation tasks. In our paper, we design the pipeline of the diffusion-based model for explanation task scenarios, as well as devise a novel classifier-guided sampling algorithm for model-level explanations.

## 3    Preliminaries

### 3.1    Problem Formulation

**Counterfactual explanation**. Given an instance (*i.e.,* a node or a graph) and a well-trained GNN, the goal of counterfactual explanation is to identify the minimal modification to the original instance that alters GNN's prediction [12, 13, 36]. Without loss of generality, we consider the explanation problem for the graph classification task. Formally, let $f$ denote a well-trained GNN classifier to be explained, $\hat{Y}_G$ denote the label of graph $G$ predicted by $f$. The counterfactual explanation $G^c$ satisfies that $\hat{Y}_{G^c} \neq \hat{Y}_G$, while the difference between $G^c$ and $G$ is minimal. This problem is usually formulated as an optimization problem that minimizes the mutual information between $G^c$ and $\hat{Y}_G$ [6, 13].

**Model-level explanation**. Model-level explanation aims to identify recurring and discriminant graph patterns that can trigger a specific prediction from the model $f$ [15, 16]. Formally, given a class $C_i \in \{C_1, \cdots, C_l\}$, model-level explanation for the target class $C_i$ can be defined as $G^m = \mathrm{argmax}_G \, P_f(C_i|G)$, where $P_f(C_i|G)$ denotes the probability for the class $C_i$ predicted by the GNN $f$, given the graph $G$. See Appendix B for more descriptions of the explanation task setting.

## 3.2 Discrete Diffusion Process for Graph

**Forward diffusion process.** In this work, we focus on discrete structural diffusion and leave the diffusion over continuous features in future work. Let $t \in [0, T]$ denote the timestep of the diffusion process, which is also a noise level indicator. Let $\boldsymbol{A}_t$ denote the one-hot version of the adjacency matrix at timestep $t$, where each element $\boldsymbol{a}_t^{ij}$ is a 2-dimensional one-hot encoding of the presence or absence of the $ij$-th element in the adjacency matrix. The forward diffusion process is a Markov chain with a transition matrix $\boldsymbol{Q}_t \in \mathbb{R}^{2 \times 2}$, that progressively transforms the input graph into pure noise. Mathematically, the forward diffusion process can be written as $q(\boldsymbol{a}_t^{ij} | \boldsymbol{a}_{t-1}^{ij}) = \text{Cat}(\boldsymbol{a}_t^{ij}; \boldsymbol{P} = \boldsymbol{a}_{t-1}^{ij} \boldsymbol{Q}_t)$, where $\text{Cat}(\boldsymbol{x}; \boldsymbol{P})$ is a categorical distribution over the one-hot vector $\boldsymbol{x}$ with probability vector $\boldsymbol{P}$. The multi-step diffusion has a closed form as $q(\boldsymbol{a}_t^{ij} | \boldsymbol{a}_0^{ij}) = \text{Cat}(\boldsymbol{a}_t^{ij}; \boldsymbol{P} = \boldsymbol{a}_0^{ij} \bar{\boldsymbol{Q}}_t)$, where $\bar{\boldsymbol{Q}}_t = \prod_{i=1}^{t} \boldsymbol{Q}_i$. See Appendix C for detailed derivation.

**Graph-level expression.** The forward diffusion process is identically and independently performed over each edge in the full adjacency matrix. Therefore, the graph-level diffusion $q(G_t | G_{t-1})$ is the product of element-wise categorical distributions as

$$q(G_t | G_{t-1}) = \prod_{ij} q(\boldsymbol{a}_t^{ij} | \boldsymbol{a}_{t-1}^{ij}) \text{ and } q(G_t | G_0) = \prod_{ij} q(\boldsymbol{a}_t^{ij} | \boldsymbol{a}_0^{ij}) \tag{1}$$

Denoising diffusion models have shown a powerful ability to recover complex distributions accurately [48, 41, 20, 44], by leveraging the diffusion process to capture intricate dependencies and generate samples that exhibit high-quality in-distribution property and diversity.

## 4 Proposed Method: D4Explainer

D4Explainer is designed for two distinct explanation scenarios: counterfactual explanation and model-level explanation. In counterfactual explanation (Sec. 4.1), D4Explainer employs a Forward diffusion process to create a sequence of noisy versions and trains a Denoising model to effectively capture the desired distribution of counterfactual graphs. For model-level explanation (Sec. 4.2), D4Explainer trains a Denoising model to recover the underlying original distribution and leverages a well-trained GNN to progressively enhance the explanation confidence during the reverse sampling. An overview is shown in Figure 2. The notation used throughout this work is summarized in Appendix A.

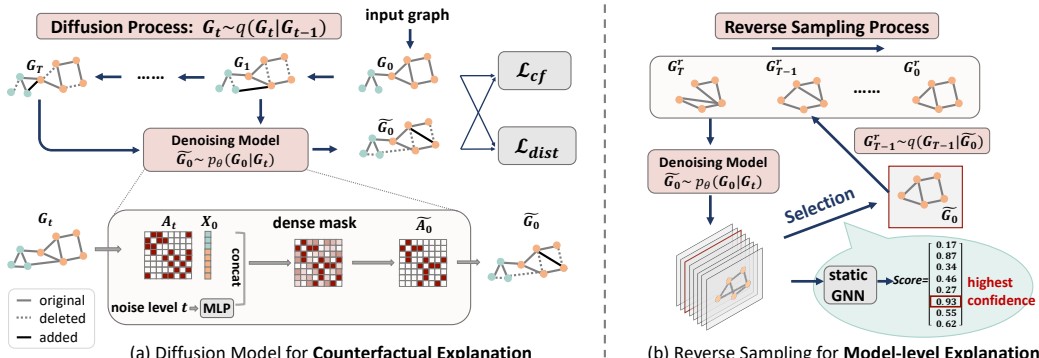

(a) Diffusion Model for **Counterfactual Explanation**          (b) Reverse Sampling for **Model-level Explanation**

Figure 2: Overview of D4Explainer. (a) Diffusion Model for counterfactual explanations. The diffusion process $q(G_t | G_{t-1})$ transforms an input graph $G_0$ to the pure noise $G_T$. Then the Denoising Model $p_\theta(\cdot)$ outputs the clean graph $\tilde{G}_0$ given a noisy graph $G_t$, under the constraints of the counterfactual loss $\mathcal{L}_{cf}$ and the distribution loss $\mathcal{L}_{dist}$. (b) Reverse Sampling for model-level explanations. We leverage a well-trained GNN to select a temporary graph with the highest confidence score from the candidate graphs and obtain $G_{t-1}^r$ from $G_t^r$ recursively until we achieve the final model-level explanation $G_0^r$.

## 4.1 Counterfactual Explanation Generation

**Forward diffusion process**. We build on the discrete diffusion process over graphs as introduced in Sec. 3.2. The forward Diffusion Process enables D4Explainer to optimize with a sequence of perturbed graphs $\{G_0, G_1, \cdots, G_T\}$ with increasing levels of noise, which essentially enable D4Explainer to thoroughly explore possible counterfactual explanations for the given graph.

**Denoising model**. To generate a counterfactual graph that closely resembles the input graph, the Denoising Model $p_\theta(G_0|G_t)$ takes as input the noisy adjacency matrix $A_t$ corresponding to a noisy graph $G_t$, the node features of the original graph $X_0$, noise level indicator $t$, and then predicts the dense adjacency matrix. Through sampling from the dense adjacency matrix with the reparameterization trick [49], we arrive at the discrete adjacency matrix $\tilde{A}_0$ and the corresponding explanation graph $\tilde{G}_0$. The Denoising Model is set as an extension of the provably powerful Graph Network (PPGN) [50]. To incorporate time information, an MLP module is employed to process the noise level indicator $t$ and learn time-related latent features, thereby enhancing the denoising capability. The edge features, node features, and time-related latent features are concatenated and updated by the powerful layers (PPGN). We refer to Appendix D.1 for a complete and detailed description of the PPGN used in our D4Explainer.

**Loss function**. Different from traditional graph generation tasks [20, 19, 47], counterfactual explanations necessitate both counterfactual property and proximity to the original graph. To address these challenges, we propose a specifically designed loss function that simultaneously optimizes these two properties. Instead of iteratively recovering the intermediate noisy graph $G_t$ in the traditional manner, we employ a re-weighted version of the evidence lower bound (ELBO) on the negative log-likelihood that directly reconstructs the initial distribution at $t = 0$ in our distribution-learning term $\mathcal{L}_{dist}$. The re-weighting strategy prioritizes more challenging denoising tasks at larger timesteps:

$$\mathcal{L}_{dist} = -\mathbb{E}_{q(G_0)} \sum_{t=1}^{T} \left( 1 - 2 \cdot \bar{\beta}_t + \frac{1}{T} \right) \mathbb{E}_{q(G_t|G_0)} \log p_\theta \left( G_0 \mid G_t \right), \tag{2}$$

where $\bar{\beta}_t$ is the transitioning probability (the off-diagonal element in the transition matrix $\bar{Q}_t$) and $q(G_0)$ is the distribution of the training dataset. The distribution loss $\mathcal{L}_{dist}$ is equivalent to the cross-entropy loss between $G_0$ and $p_\theta(G_0|G_t)$ over the full adjacency matrix, which guarantees the proximity of generated counterfactual explanations to the original graph. To optimize the counterfactual property, we design a specific counterfactual loss $\mathcal{L}_{cf}$ as follows,

$$\mathcal{L}_{cf} = -\mathbb{E}_{q(G_0)} \mathbb{E}_{t \sim [0,T]} \mathbb{E}_{q(G_t|G_0)} \mathbb{E}_{p_\theta(\tilde{G}_0|G_t)} \log \left( 1 - f(\tilde{G}_0)[\hat{Y}_{G_0}] \right), \tag{3}$$

where $f$ is the well-trained GNN classifier, $f(\tilde{G}_0)[\hat{Y}_{G_0}]$ denotes the probability for the original label $\hat{Y}_{G_0}$ predicted by $f$, given the generated graph $\tilde{G}_0$. Our total loss function is formulated as $\mathcal{L}(\theta) = \mathcal{L}_{dist} + \alpha \mathcal{L}_{cf}$, where $\alpha$ is a hyper-parameter that balances the counterfactual and in-distribution properties. Achieving the desired counterfactual property while maintaining proximity to the true data distribution involves a trade-off. For instance, making drastic modifications to the original graph may easily alter the model's prediction, but it can also lead to an explanation that deviates significantly from the original graph. The distribution loss $\mathcal{L}_{dist}$ and the counterfactual loss $\mathcal{L}_{cf}$ together encourage the denoising model to eliminate redundant edges that are irrelevant to the counterfactual property while reconstructing the original edges to preserve the true distribution.

**Working principle of D4Explainer**. D4Explainer not only preserves the **in-distribution** property but also introduces **diversity** and **robustness** to the generated counterfactual explanations. **Diversity** enables the explainer to provide multiple alternative explanations for model predictions, while **robustness** ensures consistent effectiveness of the explanations even in the presence of noise. Existing explainers often optimize a singular explanation per instance, leading to overfitting on noise and bias attribution issues [51]. On the contrary, D4Explainer's objective is to search for counterfactual graphs within the distribution of the original graphs, adhering to the constraints imposed by $\mathcal{L}_{dist}$ and $\mathcal{L}_{cf}$. Through an iterative process of adding noise and removing counterfactual-irrelevant edges, D4Explainer captures the underlying distribution of counterfactual explanations. This denoising strategy also enhances the **robustness** of D4Explainer. Moreover, the inherent stochasticity in the forward processes introduces **diversity** into the generated explanations.

## 4.2 Model-level Explanation

**Motivation**. The goal of model-level explanation is to generate class-wise graph patterns. Let $C$ denote the target class. Each reverse sampling step $q_C(G_{t-1}^r | G_t^r)$ can be formulated as a conditional generation satisfying the following equation,

$$q_C(G_{t-1}^r | G_t^r) \propto p_\theta(\tilde{G}_0 | G_t^r) q(G_{t-1}^r | \tilde{G}_0) f(C | \tilde{G}_0), \tag{4}$$

where $f(C|\tilde{G}_0)$ can be computed by the target class probability predicted by the well-trained GNN $f$, conditioned on the given graph $\tilde{G}_0$. Existing sampling methods [20, 41] cannot perform conditional sampling in the discrete context, as we cannot sample all possible $\tilde{G}_0$ to obtain $f(C|\tilde{G}_0)$ and then compute the normalized probabilities. To overcome these challenges, we propose to utilize the well-trained GNN as guidance toward the target class. At each step, we generate a set of candidates by $p_\theta(\tilde{G}_0 | G_t^r)$ and refer to the GNN to select a temporarily optimal $\tilde{G}_0$ with the highest $f(C|\tilde{G}_0)$.

**Multi-step sampling**. We repeat the sampling steps and progressively increase the explanation confidence (i.e., $f(C|\tilde{G}_0)$) in the process. Figure 3 shows an empirical visualization of the reverse generation process for the *house* motif. We observe that the temporary graph $\tilde{G}_0$ gets closer to the target motif with increasing explanation confidence $p$ during the reverse sampling process.

The proposed model-level explanation generation utilizes a denoising model trained with a similar procedure as Sec. 4.1 (Figure 2(a)). The difference is that the training loss is only $\mathcal{L}_{dist}$, since $\mathcal{L}_{cf}$ leads to a counterfactual graph that changes the label. To start with, given a pre-defined number of nodes $N$ in the target explanation, we randomly sample an Erdős–Rényi graph with $N$ nodes and edge probability $\frac{1}{2}$ as $G_T^r \sim \mathcal{B}_{N,1/2}$. Then we sample a set of candidates from the distribution $p_\theta(G_0 | G_T^r)$. The well-trained GNN computes the explanation confidences for these candidates and selects the temporary explanation $\tilde{G}_0$ with the highest score.

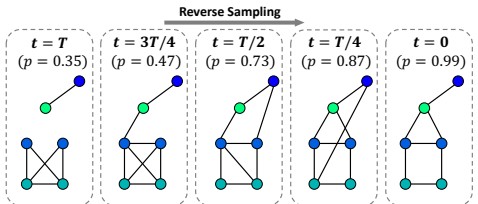

Figure 3: Visualization of the temporary $\tilde{G}_0$ at $t = T; 3T/4; T/2; T/4$ and the terminal model-level explanation for BA-3Motif (*house* motif). Different node colors indicate different labels.

Then, we sample $G_{T-1}^r$ through the same Diffusion Process $G_{T-1}^r \sim q(G_{T-1} | \tilde{G}_0)$ as Equation 1. Sampling steps iteratively reverse the chain until we obtain the final model-level explanation $G_0^r$ after $T$ steps. Apart from explanation confidence $f(C|\tilde{G}_0)$, model-level explanations should also satisfy sparsity and succinctness. It is worth noting that the proposed algorithm is capable of preserving the sparsity level similar to the training graphs in the generated explanations. For real-world datasets that are densely self-connected, it is suggested to plug regularization constraints in the selection policy for the temporary explanation at each step. The complete sampling algorithm is shown in Appendix D.4.

### 4.3 Complexity Analysis

D4Explainer has a search space of $\mathcal{O}(N^{2K})$ for modifying $K$ edges in an $N$-nodes graph, which is larger than previous counterfactual explainers that only consider deleting edges. By framing the explanation task as a generation problem, the space complexity of each layer in D4Explainer is reduced to $\mathcal{O}(N^2)$. The time complexity is $\mathcal{O}(N^3)$ due to the matrix multiplication. Despite the large search space, the complexity of D4Explainer is still acceptable and faster than some generation-based explanations [10]. Runtime and more complexity analysis are given in Appendix E.6. Furthermore, we directly recover the terminal explanation $\tilde{G}_0$ in the training procedure, rather than intermediate $G_t$, which greatly increases the efficiency of D4Explainer. The Denoising Model can also be trained in parallel under different noise levels without iterative optimization from $t = 0$ to $t = T$.

## 5 Experiments

### 5.1 Experimental Setup

We test the proposed approach to explain the performance of node classification models and graph classification models. Dataset statistics and classifier information are summarized in Appendix E.1.

Table 1: CF-ACC AUC and Fidelity (FID) AUC of D4Explainer and baseline explainers over eight datasets. We report AUC values computed over 10 modification ratios from 0 to 0.3. The best result is in **bold** and the second best result is underlined.

| Models | BA-Shapes | | Tree-Cycle | | Tree-Grids | | Cornell | | BA-3Motif | | Mutag | | BBBP | | NCI1 | |
|---|---|---|---|---|---|---|---|---|---|---|---|---|---|---|---|---|
| | CF-ACC | FID | CF-ACC | FID | CF-ACC | FID | CF-ACC | FID | CF-ACC | FID | CF-ACC | FID | CF-ACC | FID | CF-ACC | FID |
| Random | 0.251 | 0.261 | 0.260 | 0.281 | 0.337 | 0.375 | 0.138 | 0.172 | 0.404 | 0.452 | 0.192 | 0.256 | 0.073 | 0.113 | 0.288 | 0.352 |
| GNNExplainer | 0.473 | 0.444 | 0.652 | 0.580 | 0.672 | 0.622 | 0.075 | 0.120 | 0.250 | 0.253 | 0.450 | 0.449 | 0.212 | 0.241 | 0.375 | 0.443 |
| SAExplainer | 0.773 | 0.773 | 0.405 | 0.408 | 0.547 | 0.542 | 0.199 | 0.241 | 0.474 | 0.500 | 0.300 | 0.338 | 0.110 | 0.133 | 0.421 | 0.446 |
| GradCam | 0.552 | 0.570 | 0.637 | 0.613 | 0.590 | 0.578 | 0.138 | 0.189 | 0.459 | 0.495 | 0.202 | 0.250 | 0.274 | 0.301 | 0.467 | 0.488 |
| IGExplainer | 0.208 | 0.240 | 0.198 | 0.226 | 0.308 | 0.372 | 0.233 | 0.281 | 0.440 | 0.474 | 0.231 | 0.280 | 0.159 | 0.183 | 0.347 | 0.389 |
| PGExplainer | 0.361 | 0.357 | 0.353 | 0.322 | 0.293 | 0.340 | 0.128 | 0.204 | 0.320 | 0.323 | 0.208 | 0.313 | 0.233 | 0.282 | 0.338 | 0.366 |
| PGMExplainer | 0.208 | 0.210 | 0.242 | 0.214 | 0.128 | 0.237 | 0.206 | 0.274 | 0.212 | 0.213 | 0.128 | 0.251 | 0.105 | 0.154 | 0.348 | 0.390 |
| CXPlain | 0.125 | 0.168 | 0.245 | 0.220 | 0.222 | 0.274 | 0.132 | 0.180 | 0.235 | 0.239 | 0.187 | 0.305 | 0.067 | 0.131 | 0.489 | 0.484 |
| CF-GNNExplainer | 0.773 | 0.728 | 0.812 | 0.718 | 0.537 | 0.527 | 0.328 | 0.297 | 0.302 | 0.304 | **0.797** | **0.751** | 0.623 | 0.632 | 0.715 | 0.674 |
| **D4Explainer** | **0.838** | **0.828** | **0.917** | **0.862** | **0.905** | **0.832** | **0.623** | **0.559** | **0.912** | **0.922** | 0.765 | 0.675 | **0.781** | **0.739** | **0.737** | **0.690** |

**Node classification**. For synthetic datasets, we use BA-Shapes, Tree-Cycle, Tree-Grids [6]. There exists a motif that plays an important role in the model's prediction. The node labels are determined by the structural roles. We train a vanilla GCN for synthetic datasets, achieving over $95\%$ accuracy on each synthetic dataset. Additionally, we use Cornell [52] dataset, a highly heterophilous real-world webpage graph. Wherein, more complex relationships exist between a node and its neighbors, thus posing a more significant challenge to the explanation tasks. We train an EGNN [53], which is specifically designed for heterophilous graphs, achieving $83\%$ accuracy on Cornell.

**Graph classification**. We use one synthetic dataset, BA-3Motif [54] and three real-world molecule datasets, Mutag [55, 56], BBBP [57] and NCI1 [58] for graph-classification task explanation. BA-3Motif contains 3 graph classes: graphs with cycle motif, grid motif, and house motif. Mutag, BBBP, and NCI1 are molecular datasets where nodes represent atoms and graphs represent molecules. Specifically, the chemical functionalities of molecules determine the graph labels. We train a vanilla GCN for BA-3Motif, BBBP, and NCI1. For the Mutag dataset, GIN [59] is used as the target GNN.

**Baselines**. For the counterfactual explanation task, we take the same baseline setup as CF-GNNExplainer[13] and involve more recent state-of-the-art explainers as our baselines, including GNNExplainer [6], SAExplainer [29], GradCam [21], IGExplainer [22], PGExplainer [8], PGMExplainer [9], and CXPlain [60]. For the methods that are originally designed for the factual explanation, we construct a subgraph with the least important edges as the counterfactual explanation. For the model-level explanation task, we compare with XGNN [15], which is a state-of-the-art model-level explanation method for GNNs. More implementation details are given in Appendix E.3.

## 5.2 Counterfactual Explanations

**Metrics**. Following evaluation protocols of prior works [13, 36], we adopt Counterfactual Accuracy, Fidelity, and Modification Ratio (MR) as our metrics. Let $G^o$ and $G^c$ denote the original input graph and generated counterfactual graph, respectively. $\mathcal{G}$ is the test dataset. Counterfactual Accuracy is defined as the proportion of generated explanations that change the model's prediction, CF-ACC $= 1 - 1/|\mathcal{G}| \sum_{G^o \in \mathcal{G}} (\mathbb{1}(\hat{Y}_{G^c} = \hat{Y}_{G^o}))$. Fidelity measures the change in output probability over the original class, *i.e.,* Fidelity $= 1/|\mathcal{G}| \sum_{G^o \in \mathcal{G}} \left[ f(G^o)[\hat{Y}_{G^o}] - f(G^c)[\hat{Y}_{G^o}] \right]$. Modification Ratio refers to the proportion of changed edges as MR $=$ (# of deleted edges + # of added edges)$/|E|$. Higher counterfactual accuracy and fidelity with lower modification ratios indicate better performance.

**Results**. CF-ACC and Fidelity are sensitive to the modification ratio, we thus compute the areas under CF-ACC curve and Fidelity curve over 10 different modification ratios from 0 to 0.3. We run 10 different seeds for each approach and report the average in Table 1. As can be seen from the table, D4Explainer achieves the best performances on seven out of eight datasets, with especially strong CF-ACC AUC values ($> 90\%$) on Tree-Cycle, Tree-Grids, and BA-3Motif. Notably, D4Explainer consistently works well on explaining both node classification and graph classification tasks, while the efficacy of baselines is unstable across datasets. For instance, most baselines fail to generate effective counterfactual explanations for complex graphs with multiple motifs or heterophilous edge relations, *e.g.,* Cornell and BA-3Motif.

To further investigate the relation between CF-ACC and the modification ratio, we show the change of CF-ACC *w.r.t.* modification ratios from 0 to 0.3 in Figure 4, where the X-axis is in the log scale.

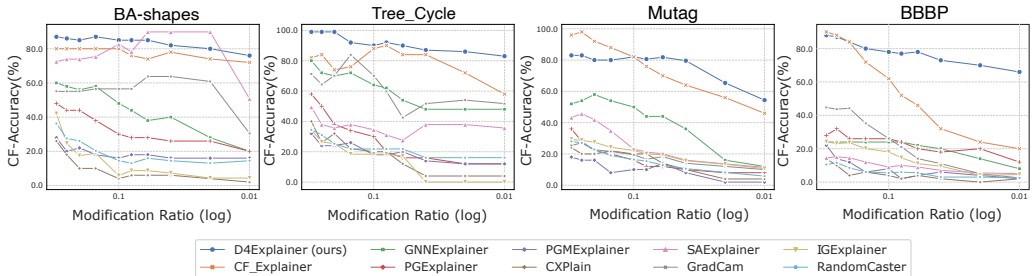

Figure 4: `CF-ACC` Curves of all explainers over different modification ratios from $0$ to $0.3$. The $x$-axis is shown in the $\log$ scale. `CF-ACC` tends to increase as the modification ratio increases in general.

Table 2: MMD distances between the generated explanations and test graphs. We report MMD distances of degree distributions (*Deg.*), cluster coefficients (*Clus.*), spectrum distributions (*Spec.*), and the summation (*Sum.*). We bold the best value and underlined the second-best value.

| Models | Mutag | | | | BBBP | | | | NCI1 | | | |
|---|---|---|---|---|---|---|---|---|---|---|---|---|
| | Deg. | Clus. | Spec. | Sum. | Deg. | Clus. | Spec. | Sum. | Deg. | Clus. | Spec. | Sum. |
| RamdomCaster | 0.1593 | 0.0247 | 0.0417 | 0.2257 | 0.1693 | 0.0072 | 0.0397 | 0.2162 | 0.1847 | 1.9769 | 0.0404 | 2.2020 |
| GNNExplainer | 0.1614 | 0.0002 | 0.0409 | 0.2025 | 0.1615 | 0.0002 | 0.0395 | 0.2012 | 0.1577 | 0.0005 | 0.0405 | 0.1987 |
| SAExplainer | **0.0940** | 0.0032 | 0.0412 | **0.1384** | 0.1594 | 0.0032 | 0.0402 | 0.2028 | 0.189 | 0.0002 | 0.0408 | 0.2300 |
| GradCam | 0.1122 | 0.0083 | 0.0416 | 0.1621 | 0.0699 | 0.0026 | 0.0384 | 0.1109 | 0.1638 | 0.0003 | 0.0404 | 0.2045 |
| IGExplainer | 0.1292 | **0.0000** | 0.0411 | 0.1703 | 0.0908 | **0.0000** | 0.0394 | 0.1302 | 0.4288 | 0.0002 | 0.0398 | 0.4688 |
| PGExplainer | 0.1475 | 0.0002 | 0.0418 | 0.1895 | 0.2014 | 0.0018 | 0.0403 | 0.2435 | 0.1937 | **0.0000** | 0.0396 | 0.2333 |
| PGMExplainer | 0.1800 | 0.0002 | 0.0419 | 0.2221 | 0.1916 | 0.0003 | 0.0403 | 0.2322 | 0.2199 | **0.0000** | 0.0404 | 0.2603 |
| CXPlain | 0.1734 | 1.2706 | 0.0417 | 1.4857 | 0.1768 | 0.0001 | 0.0394 | 0.2163 | 0.1629 | 0.0001 | 0.0404 | 0.2034 |
| CF-GNNExplainer | 0.1172 | **0.0000** | 0.0380 | 0.1552 | 0.0870 | 0.0001 | 0.0393 | 0.1264 | 0.1224 | 0.0001 | 0.0404 | 0.1629 |
| **D4Explainer** | 0.1172 | **0.0000** | **0.0244** | 0.1416 | **0.0530** | **0.0000** | **0.0331** | **0.0861** | **0.1006** | **0.0000** | **0.0353** | **0.1359** |

As illustrated in Figure 4, D4Explainer consistently achieves the highest `CF-ACC` with the smallest modification ratio (see the right side of the X-axis). Especially for Tree-Cycle and BBBP dataset, D4Explainer obtains a significant boost compared to the baselines. It demonstrates that D4Explainer can generate counterfactual explanations that can strongly influence the prediction of the target GNN and reflect the effective counterfactual properties.

### 5.2.1 In-Distribution Evaluation

To evaluate the in-distribution property of the generated explanations, we adopt the maximum mean discrepancy (MMD) to compare distributions of graph statistics between the generated counterfactual explanations and original test graphs. Following the evaluation setting in prior works [61, 19, 20, 62, 44], we use Gaussian Earth Mover's Distance kernel to compute MMDs of degree distributions, clustering coefficients, and spectrum distributions. Smaller MMDs mean that the two distributions are more similar and close, which indicates a better in-distribution property.

**Results**. Table 2 shows the MMD results on three real-world molecular datasets. We observe that D4Explainer outperforms baselines in general. Especially for BBBP and NCI1 datasets, D4Explainer achieves the lowest MMD distances across all metrics. The MMD results verify the effectiveness of D4Explainer in capturing the underlying distribution of datasets and generating in-distribution and more faithful explanations. We refer to Appendix E.4 for more results.

### 5.2.2 Additional Faithfulness Aspects

**Explanation Diversity Evaluation.** We evaluate the diversity of counterfactual explanations in Figure 5 and Appendix E.5. The first row shows the original graphs. The second row shows the generated counterfactual explanations by CF-GNNExplainer [6], where only edge deletion is allowed. With edge addition, D4Explainer is capable of generating alternative counterfactual explanations from a different perspective. As can be found from Figure 5, there are two main approaches to generating counterfactual explanations. The first one is deleting determinant edges and destroying the original motif, thus greatly influencing the model's prediction. The second one is converting the original motifs to truly counterfactual motifs through both deleting and adding essential edges. Previous methods can only produce the first type of counterfactual explanations, while D4Explainer makes the second approach possible and successful, leading to alternative and diverse counterfactual

explanations. We ascribe the success to the special training mechanism of D4Explainer. The intrinsic stochasticity in the forward process allows D4Explainer to take as input a sequence of noisy versions of the original graph, instead of a singular input graph. This enlarges the search space of possible counterfactual explanations for D4Explainer.

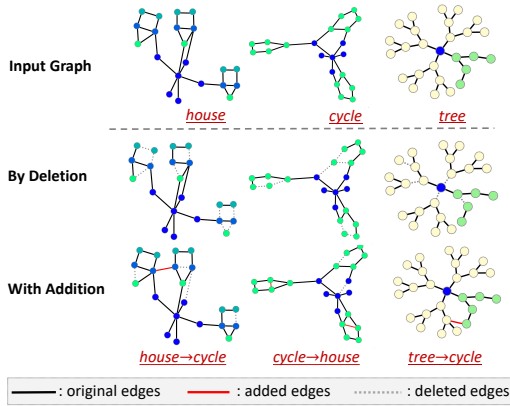

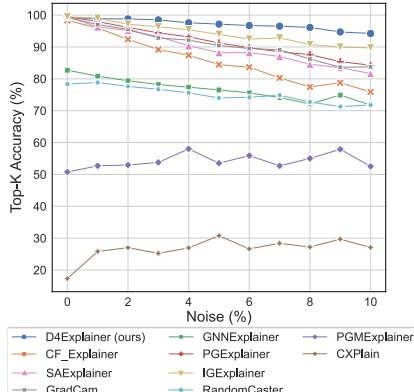

Figure 5: Counterfactual explanations comparison. Red labels represent the motifs in the graph.

Figure 6: Top-K accuracy *w.r.t.* noise levels on BBBP dataset

**Robustness Evaluation.** To evaluate the robustness of all methods, we compare the counterfactual explanations produced on the original graph and its perturbed counterpart, respectively. A robust model would predict the same explanation for both inputs. Following previous setup [36], we identify the $K$ most relevant edges in the original counterfactual explanation and compute the fraction of these edges present in the explanation of its noisy version, denoted by Top-$K$ Accuracy. We apply noise by randomly adding or removing edges with probability $\sigma$. A consistent $20\%$ modification ratio is used across all methods. Results on BBBP dataset are shown in Figure 6. We observe that D4Explainer outperforms all baselines over different noise levels from 0 to $10\%$. We restrict that $\sigma < 10\%$, as the larger noise may cause the noisy graph to switch the predicted label. Overall, results in Figure 6 verify D4Explainer's strong ability to generate consistently effective counterfactual explanations despite the noise. See Appendix E.7 for complete results.

## 5.3 Model-level Explanations

In each step of the reverse sampling, we denoise $K$ candidate graphs from the noisy graph and select a temporary explanation. Following the setting in XGNN [15], we qualitatively evaluate the generated explanations with different pre-defined numbers of nodes $N$, shown in Figure 7. $p$ denotes the target class probability predicted by the GNN. A higher $p$ indicates higher explanation confidence. We observe that D4Explainer can produce more determinant graph patterns with nearly $100\%$ confidence for synthetic datasets, *e.g.,* BA-shapes and BA-3Motif.

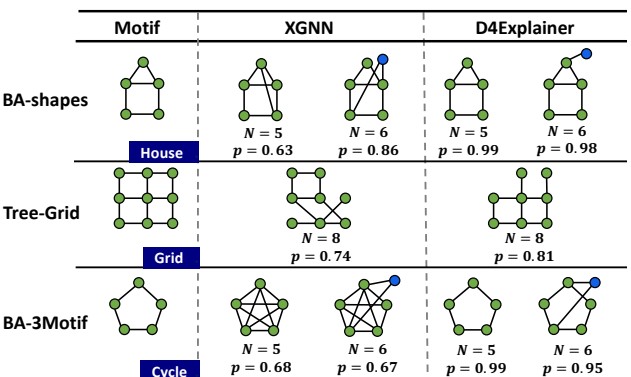

Figure 7: Qualitative evaluation of D4Explainer

**Quantitative evaluation**. We adopt the target class probability $p$ and `Density` as the quantitative metrics. Density measures the sparsity level of the explanations, which is defined as $\texttt{Density} = |\mathcal{E}|/|\mathcal{V}|^2$, where $\mathcal{E}$ and $\mathcal{V}$ denote the set of edges and nodes in the explanation. Quantitative comparisons between XGNN and D4Explainer under different numbers of nodes are shown in Table 3. Hyperparameter sensitivities of the $K$ (number of candidates in each step) and $T$ (number of reverse sampling steps) are shown in Table 4. The results are averaged over 100 generated model-level explanations without

any regularization constraints in the selection policy. We find that (1) D4Explainer is capable of generating sparse and succinct model-level explanations with high target class probabilities, even without any regularization constraints on the explanation size. The superiority can be attributed to our distribution learning objective. However, it is worth noting that training graphs might be noisy and densely self-connected in some real-world applications. A regularization constraint can be easily plugged into the selection policy if required by downstream tasks; (2) smaller $T$ and $K$ both degrade the performance and quality of model-level explanations, which further emphasize the effectiveness of candidates and multi-step sampling. In the implementation, we ensure $K \geqslant 20$ and $T \geqslant 50$ for a balance between the quality and time complexity.

Table 3: Quantitative comparison in terms of probability and density with different numbers of nodes.

|  |  | Mutag | | | Tree-Cycle | | |
|---|---|---|---|---|---|---|---|
|  | # nodes | 6 | 7 | 8 | 5 | 6 | 7 |
| **Ours** | Prob. | **0.832** | **0.856** | **0.920** | **0.991** | **0.995** | 0.989 |
|  | Density | **0.278** | **0.327** | **0.315** | 0.400 | **0.381** | **0.343** |
| **XGNN** | Prob. | 0.523 | 0.824 | 0.875 | 0.968 | 0.989 | **0.992** |
|  | Density | 0.537 | 0.479 | 0.437 | 0.400 | 0.390 | 0.367 |

Table 4: Hyperparameter sensitivity in model-level explanation generation

|  | Mutag (N=6) | | Tree-Cycle (N=6) | |
|---|---|---|---|---|
|  | Prob. | Density | Prob. | Density |
| (1) $K = 10, T = 50$ | 0.799 | 0.314 | 0.987 | 0.372 |
| (2) $K = 20, T = 10$ | 0.524 | 0.284 | 0.991 | 0.388 |
| (3) $K = 20, T = 50$ | 0.812 | 0.295 | 0.994 | 0.361 |
| (4) $K = 20, T = 100$ | **0.832** | **0.278** | 0.992 | **0.325** |
| (5) $K = 30, T = 50$ | 0.823 | 0.287 | **0.997** | 0.361 |

## 6 Conclusion and Broader Impacts

In this work, we propose D4Explainer, a novel generative approach for counterfactual and model-level explanations based on a discrete denoising diffusion model. By framing the explanation problem as a distribution learning task, D4Explainer can generate more reliable explanations with better in-distribution property, diversity and robustness. Additionally, D4Explainer can simultaneously perform model-level explanations with a pre-trained denoising model.

While denoising diffusion models show promise for explaining Graph Neural Networks (GNNs), they face potential scalability concerns on large graphs. Additionally, the explanations rely on the specific GNN architecture, limiting their generalizability across different GNN models. This work has dual social impacts. It enhances the transparency and interpretability of GNNs. However, it is vital to acknowledge the limitations and potential risks of relying solely on these explanations. They may not always capture the complete causal relationships in complex graph structures, which could lead to unintended consequences, reinforce biases, or make incorrect assumptions about the model's behavior. Looking ahead, an interesting direction for future research is to consider the node attributes and edge attributes during the explanation generation, *e.g.,* by performing diffusion processes over continuous features.

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

# A    Notations

The main notations used throughout this paper are summarized in Table 5.

Table 5: Summary of the notations

| Notation | Description |
|---|---|
| $G$ | An input graph / computational graph for a given node |
| $G_0$ | Original graph *i.e.,* $G$ |
| $X_0$ | Node features |
| $t$ | The timestep of the forward diffusion |
| $G_t$ | Noisy graph at timestep $t$ |
| $f$ | A well-trained GNN classifier to be explained |
| $\hat{Y}_G$ | The label of graph $G$ predicted by $f$ |
| $G^c$ | Counterfactual explanation of $G$ |
| $\{C_1, \cdots, C_l\}$ | Class set of the input graphs |
| $G^m$ | Model-level explanation for a certain class |
| $q(G_t \mid G_{t-1})$ | Forward diffusion process |
| $p_\theta(\cdot \mid G_t)$ | Denoising model |
| $\tilde{G}_0$ | Reconstructed clean graph |
| $\mathcal{L}_{cf}$ | Counterfactual loss |
| $\mathcal{L}_{dist}$ | Distribution loss |
| $G_t^r$ | Graph at timestep $t$ in the reverse sampling process |
| $p$ | Explanation confidence |
| $K$ | Number of candidates in each step for the model-level explanation |
| $T$ | Number of reverse sampling steps for the model-level explanation |
| $N$ | (Predefined) number of nodes in the target model-level explanation |

# B    Explanation Setting

The counterfactual explanation for a prediction highlights the smallest change to the original instance that changes the prediction. It is a post-hoc step after the model is designed and well-trained.

**Definition 1.** *(Counterfactual Explanation) Given a well-trained classifier $f$ that predicts the label $\hat{Y}_G$ for an instance $G$, a counterfactual explanation consists of an instance $G^c$ such that the prediction on $G^c$ is different from the original $\hat{Y}_G$ on $G$, such that the difference between $G$ and $G^c$ is minimal.*

It can be formulated as an optimization problem that minimizes the mutual information [6, 13]:

$$\underset{d(G,G^c)<K}{\operatorname{argmin}}\ MI(\hat{Y}_G, G^c) \Leftrightarrow \underset{d(G,G^c)<K}{\operatorname{argmax}}\ H(\hat{Y}_G|G^c) \Leftrightarrow \underset{d(G,G^c)<K}{\operatorname{argmin}}\ -\mathbb{E}_{\hat{Y}_G|G^c}\left[\log\left(1 - P_f(\hat{Y}_G \mid G^c)\right)\right]$$

(5)

where $MI(\cdot)$ is the mutual information function, $H(\cdot)$ is the entropy function and $H\left(\hat{Y}_G \mid G^c\right) = -\mathbb{E}_{\hat{Y}_G|G^c}\left[\log P_f\left(\hat{Y}_G \mid G^c\right)\right]$. $P_f\left(\hat{Y}_G \mid G^c\right)$ denotes the probability for the $\hat{Y}_G$ label given the counterfactual explanation $G^c$, predicted by $f$. $d(G, G^c)$ measures the proximity between the original $G$ and counterfactual explanation $G^c$, which can be specified by the number of changed edges (including the removed edges and newly added edges). An ideal counterfactual explanation should be similar to the original graph, therefore, $K$ is applied as a constraint over the proximity.

**Definition 2.** *(Model-level Explanation) Given a well-trained GNN classifier $f$ and a set of graph $\mathcal{G}$ that is predicted as the same label $C_i$ by $f$, the model-level explanation for the target class $C_i$ is a recurrent and determinant graph pattern that leads to the certain prediction made by $f$.*

Formally, the model-level explanation for the target class $C_i$ can be formulated as $G^m = \operatorname{argmax}_G P_f(C_i|G)$, where $P_f(C_i|G)$ can be computed by the probability for the class $C_i$ predicted by the well-trained GNN $f$. Meanwhile, the model-level explanations should be recurrent in the given graph set. Typically, the model-level explanations should adhere to the distribution of the input graphs to be representative of the graph characterizations [16].

## C   Diffusion Process

### C.1   Discrete Diffusion Process for Graph

**Forward diffusion process**. Let $t \in [0, T]$ denote the timestep of the diffusion process, which is also a noise level indicator. Let $A_t$ denote the one-hot version of the adjacency matrix at timestep $t$, where the row vector $a_t^{ij} \in \{0, 1\}^2$ is a 2-dimensional one-hot encoding of the $ij$-th element $a_t^{ij}$ in the adjacency matrix $A_t$. The *forward diffusion process* is a Markov chain that progressively transforms the input graph into pure noise. The forward transition probabilities can be represented by a transition matrix $Q_t \in \mathbb{R}^{2 \times 2}$, where the $rs$-th element $Q_t^{rs} = q(a_t^{ij} = s | a_{t-1}^{ij} = r)$. For example, $Q_t^{01}$ indicates the probability of being absent at timestep $t-1$ and transitioning to being present at timestep $t$ for each edge. Therefore, the transition matrix $Q_t$ can be represented as

$$Q_t = \begin{pmatrix} 1 - \beta_t & \beta_t \\ \beta_t & 1 - \beta_t \end{pmatrix}, \tag{6}$$

where $1 - \beta_t$ models the probability that an edge state does not change at timestep $t$ (e.g., remaining present or remaining absent in the graph). With the transition matrix and one-hot encoding $a_t^{ij}$, the forward diffusion process can be written as $q(a_t^{ij} | a_{t-1}^{ij}) = \text{Cat}(a_t^{ij}; P = a_{t-1}^{ij} Q_t)$, where $\text{Cat}(x; P)$ is a categorical distribution over the one-hot vector $x$ with probability vector $P$.

**Multi-step diffusion**. The formulation of Equation 6 allows for computing multiple-step diffusion from $a_0^{ij}$ to $a_t^{ij}$ directly in a closed form, by $q(a_t^{ij} | a_0^{ij}) = \text{Cat}(a_t^{ij}; P = a_0^{ij} \bar{Q}_t)$, where $\bar{Q}_t = \prod_{i=1}^{t} Q_i$. Additionally, $\bar{Q}_t$ can also be represented as a symmetric matrix like Equation 6, with $\beta_t$ being replaced by

$$\bar{\beta}_t = \frac{1}{2} - \frac{1}{2} \prod_{i=1}^{t} (1 - 2\beta_i). \tag{7}$$

In the implementation, we only perform multi-step diffusion. We uniformly sample $\bar{\beta}_t$ in the range of $[0, 0.5]$ to control the level of noise.

**Graph-level expression**. The forward diffusion process is independently performed over all of the edges in the full adjacency matrix. Therefore, the graph-level diffusion $q(G_t | G_{t-1})$ is the product of element-wise categorical distributions as

$$q(G_t | G_{t-1}) = \prod_{ij} q(a_t^{ij} | a_{t-1}^{ij}) \text{ and } q(G_t | G_0) = \prod_{ij} q(a_t^{ij} | a_0^{ij}) \tag{8}$$

The forward diffusion process transforms the input graph into pure noise when $T$ goes to infinity. The pure noise graph $G_\infty$ is an Erdős–Rényi random graph [63] with the probability $\frac{1}{2}$ of being present or absent for each edge.

### C.2   Continuous Diffusion Process

In this section, we discuss the extension of D4Explainer for the diffusion over continuous content features (*i.e.,* node features, edge features, etc). Given a graph $G(X_0, A_0)$ with the initial node features $X_0 \in \mathbb{R}^{N \times d}$ and initial adjacency matrix $A_0 \in \{0, 1\}^{N \times N}$, where $N, d$ is the number of nodes and feature dimensions respectively. Let $X_t, A_t$ denote the noisy node feature and noisy adjacency matrix at timestep $t$. $A_t$ is obtained by the discrete diffusion process in Sec. 3.2, while continuous noisy node features $X_t$ rely on continuous Gaussian perturbations. The forward Markov process gradually adds Gaussian noise to the previous state:

$$q(X_t | X_{t-1}) = \mathcal{N}(X_t; \sqrt{1 - \beta_t} X_{t-1}, \beta_t \mathbf{I}), \tag{9}$$

where $\mathcal{N}$ denotes the high-dimensional Gaussian distribution, $\beta_t$ is the variance at timestep $t$. Similar to discrete diffusion, there is a closed form that performs multi-step diffusion:

$$q(X_t | X_0) = \mathcal{N}(X_t; \sqrt{\bar{\alpha}_t} X_0; (1 - \bar{\alpha}_t) \mathbf{I}), \tag{10}$$

where $\alpha_t = 1 - \beta_t$ and $\bar{\alpha}_t = \prod_{s=1}^{t} \alpha_s$. The denoising model takes as input the adjacency matrix $A_t$, the noisy node features $X_t$ and the noisy level indicator $t$ and predicts the clean adjacency matrix $\tilde{A}_0$

and node features $\tilde{X}_0$. Let $\tilde{G}_0(\tilde{X}_0, \tilde{A}_0)$ denote the predicted explanatory graph. With the continuous diffusion over node features, we need to recover both the original adjacency matrix and original node features. Thus it becomes the cross-entropy between $A_0$ and $\tilde{A}_0$ as well as the cross-entropy between $X_0$ and $\tilde{X}_0$. Therefore, the distribution loss can be expressed as

$$
\begin{aligned}
\mathcal{L}_{dist} = &-\mathbb{E}_{q(A_0)} \sum_{t=1}^{T} \left(1 - 2 \cdot \bar{\beta}_t + \frac{1}{T}\right) \mathbb{E}_{q(A_t|A_0)} \log p_\theta\left(A_0|A_t\right) \\
&- \mathbb{E}_{q(X_0)} \sum_{t=1}^{T} \left(1 - 2 \cdot \bar{\beta}_t + \frac{1}{T}\right) \mathbb{E}_{q(X_t|X_0)} \log p_\theta(X_0|X_t)
\end{aligned}
\tag{11}
$$

The above continuous setting can also easily generalize to edge features diffusion.

## D   Model Details

### D.1   Denoising Model: PPGN

Our PPGN implementation follows the original paper [50], and [20]. The difference is that we insert an MLP module that processes the noise level indicator $t$ and learns time-related latent features to enhance the denoising capability. Given a graph $G$, let $\boldsymbol{A}_t \in \mathbb{R}^{N \times N \times 2}$ denote the one-hot version of the adjacency matrix at timestep $t$, where the row vector $\boldsymbol{a}_t^{i,j} \in \{0,1\}^2$ is a 2-dimensional one-hot encoding of the existence of the edge between node $i$ and node $j$, $N$ is the number of nodes in the graph. Let $X = [X_1, \cdots, X_N] \in \mathbb{R}^{N \times d}$ denote the node features of the original graph, where $d$ denotes the number of feature dimensions, $X_i \in \mathbb{R}^d$ denotes the $d$-dimensional feature of the node $i$. We construct $\boldsymbol{X} \in \mathbb{R}^{N \times N \times 2d}$, where $\boldsymbol{X}_{ij} \in \mathbb{R}^{2d}$ is the concatenation of node feature $X_i$ and $X_j$. Specifically, we use a diagonal matrix $\bar{\beta}_t \cdot \boldsymbol{I} \in \mathbb{R}^{N \times N \times 1}$ as the noise level indicator. An MLP module will process the time-related information and output a tensor $\text{MLP}(\bar{\beta}_t \cdot \boldsymbol{I}) \in \mathbb{R}^{N \times N \times 1}$

Let $\boldsymbol{M}_{in} = \text{Cat}(\boldsymbol{A}_t, \boldsymbol{X}, \text{MLP}(\bar{\beta}_t \cdot \boldsymbol{I})) \in \mathbb{R}^{N \times N \times (2d+3)}$ as the input of PPGN model. The output tensor of PPGN is $A_0' \in \mathbb{R}^{N \times N \times 1}$, where each element $[A_0']_{ij}$ represents the probability of $q(\boldsymbol{a}_t^{ij}|\boldsymbol{a}_0^{ij})$. The formulation of PPGN is as follows,

$$
\begin{aligned}
PGNN(\boldsymbol{M}_{in}) &= L_{out} \circ C(\boldsymbol{M}_{in}) \\
C(\boldsymbol{M}_{in}) &= \text{Concat}((B_d \circ \cdots B_1)(\boldsymbol{M}_{in}), \\
&\quad (B_{d-1} \circ \cdots B_1)(\boldsymbol{M}_{in}), \cdots, B_1(\boldsymbol{M}_{in})) \in \mathbb{R}^{N \times N \times (dh)}
\end{aligned}
\tag{12}
$$

Each $B_i$ is a powerful layer that maps the input tensor to a tensor in $\mathbb{R}^{N \times N \times h}$. We concatenate $d$ outputs of these powerful layers and obtain a tensor $C(\boldsymbol{M}_{in}) \in \mathbb{R}^{N \times N \times (dh)}$. The final $L_{out}$ is an MLP module that maps the input tensor to the space of $\mathbb{R}^{N \times N \times 1}$: $L_{out} : \mathbb{R}^{N \times N \times (dh)} \rightarrow \mathbb{R}^{N \times N \times 1}$. We take the output of the PPGN model as the dense adjacency matrix as mentioned in Sec. 4.1.

### D.2   Counterfactual Explanation Generation

The output of PPGN model $A_0' = PGNN(\boldsymbol{M}_{in}) \in \mathbb{R}^{N \times N \times 1}$ is taken as the dense adjacency matrix for the counterfactual explanation, where each element indicates the probability of the corresponding edge in the final counterfactual explanation. To obtain the discrete adjacency matrix and backpropagate the gradients, we utilize the Concrete relaxation of the Bernoulli distribution via

$$
\texttt{Bernoulli}(p) \approx \sigma(\frac{1}{\lambda}(\log p - \log(1-p) + \log u - \log(1-u))), \quad u \sim \texttt{Uniform}(0,1),
$$

where $\lambda$ is a temperature for the Concrete distribution and $\sigma$ is the sigmoid function. Then, we create a discrete adjacency matrix by $\tilde{A}_0[ij] \sim \texttt{Bernoulli}(A_0'[ij])$, where $[ij]$ denotes the $ij$-th element in the corresponding matrix. Once the denoising model is well trained, we can generate a counterfactual explanation given any noisy graph $G_t$, the node feature $\boldsymbol{X}$, and noisy indicator $t$. In the explanation stage, let $G_0$ denote the given graph to be explained, we randomly add noise to $G_0$ and create a noisy version. We utilize the well-trained denoising model to output a dense adjacency matrix $A_0'$. The reparametrization trick is not applied in the inference stage. We directly sample $\tilde{A}_0[ij] \sim \texttt{Bernoulli}(A_0'[ij])$ and construct the final counterfactual explanation. One may also calculate average $A_0'$ by denoising from multiple noisy versions $G_t$ with different noisy level indicators $t$.

### D.3 Simplified Loss Function

Early efforts on denoising diffusion models mainly reconstruct each $G_{t-1}$ from $G_t$. However, it poses a challenge to the training stability due to the dependence of $G_{t-1}$ on the sampled diffusion trajectories and the intrinsic noise of $G_{t-1}$. The simplified loss was first proposed by [48], which is defined as

$$\mathcal{L}_{simple} = -\mathbb{E}_{q(G_0)}\mathbb{E}_{t \sim [0,T]}\mathbb{E}_{q(G_t|G_0)} \log p_\theta\left(G_0 \mid G_t\right).$$

Instead of reconstructing intermediate noisy graphs, the simplified loss directly pushes toward the terminal clean graph $G_0$, which improves both the training stability and training efficiency. In this work, we also target at recovering the final counterfactual graphs $\tilde{G}_0$ with each noisy graph $G_t$. Moreover, we emphasize more challenging denoising tasks at larger timesteps by adding the weight $1 - 2 \cdot \bar{\beta}_t + \frac{1}{T}$ to each step.

### D.4 Model-level Explanation Generation

---
**Algorithm 1** Reverse Sampling for Model-level Explanation

---
**Require:** number of nodes $N$, number of candidates $K$, static GNN $f$, Diffusion Process $q(\cdot)$, Denoising Model $p_\theta(\cdot)$
1: Sample an Erdős–Rényi graph $G_T^r \sim \mathcal{B}_{N,1/2}$
2: **for** $t = T$ to 1 **do**
3:     Sample candidates$\{\tilde{G}_{0,k} \mid \tilde{G}_{0,k} \sim p_\theta(G_0|G_t^r); k = 1, \cdots, K\}$
4:     Compute $Prob[k] = f(\tilde{G}_{0,k})$ for $k = 1, \cdots, K$
5:     Select $\tilde{G}_{0,j}$ with the highest $Prob[j]$
6:     Temporary explanation $\tilde{G}_0 := \tilde{G}_{0,j}$
7:     Sample $G_{t-1}^r \sim q(G_{t-1}|\tilde{G}_0)$
8: **end for**
9: **return** $G_0^r$

---

Alg. 1 shows the multi-step reverse sampling algorithm for model-level explanations. Let $N$ denote the number of nodes in the desired model-level explanation. We first generate a pure random graph $G_T^r \sim \mathcal{B}_{N,1/2}$. Given the noisy graph $G_T^r$, the denoising model predicts the distribution of the clean graphs by $p_\theta(G_0|G_T^r)$. We sample $K$ candidates from the distribution of the clean graphs by $\tilde{G}_{0,k} \sim p_\theta(G_0|G_T^r)$, with $k = 1, \cdots, K$, and refer to the well-trained GNN to select the optimal one with the highest explanation confidence (*i.e.,* $f(C_i|\tilde{G}_{0,k})$). Regularization constraints can be plugged into this step to further guarantee the desired properties of the generated explanation [16], *e.g.,* sparsity, explanation size, connectivity incentive, *etc.* We nominate the optimal $\tilde{G}_{0,j}$ as the temporary explanation $\tilde{G}_0$. Then, $\tilde{G}_0$ is transformed to noisy graphs by forward diffusion process, *i.e.,* $G_{t-1}^r \sim q(G_{t-1}|\tilde{G}_0)$. We repeat the process for $T$ times until we obtain the terminal $G_0^r$ as the model-level explanation.

### D.5 Unification of D4Explainer

The unification of D4Explainer lies in the same diffusion process and denoising model for different explanation scenarios. The differences between D4Explainer on counterfactual and model-level explanation tasks are (1) loss function and (2) reverse sampling process. Specifically, the loss function for the model-level explanation task does not contain $\mathcal{L}_{cf}$, which is designed to ensure the counterfactual property. Moreover, the reverse sampling process in the model-level explanation tasks utilizes multiple-step sampling to increase the explanation confidence score of generated model-level explanations. Moreover, the flexibility in the loss function and reverse sampling process enable D4Explainer to tackle other related explanation scenarios, such as instance-level factual explanation.

Table 6: Statistics of the eight datasets and test performance of the GNN model trained for each dataset. "-" means that there is no ground-truth motif in the dataset

|  | BA-Shapes | Tree-Cycle | Tree-Grids | Cornell | BA-3Motif | Mutag | BBBP | NCI1 |
|---|---|---|---|---|---|---|---|---|
| # of Nodes (avg.) | 700 | 871 | 1231 | 183 | 21.92 | 30.32 | 25.95 | 29.87 |
| # of Edges (avg.) | 4110 | 1942 | 3130 | 280 | 29.51 | 30.77 | 24.06 | 32.30 |
| # of Graphs | 1 | 1 | 1 | 1 | 3000 | 4337 | 2039 | 4110 |
| # of Classes | 4 | 2 | 2 | 5 | 3 | 2 | 2 | 2 |
| Motif | house | cycle | grid | - | house/cycle/grid | - | - | - |
| Target GNN | GCN | GCN | GCN | EGNN | GCN | GIN | GCN | GCN |
| Test accuracy | 0.99 | 0.98 | 0.95 | 0.83 | 0.93 | 0.87 | 0.85 | 0.83 |

# E    Experiments

## E.1    Dataset

In this work, we use four synthetic datasets: BA-shapes, Tree-Cycle, Tree-Grids, and BA-3Motif to evaluate the efficacy of the proposed D4Explainer . In the node-classification task, the graph consists of a base graph, which is randomly attached by different motifs, *e.g., house*, *grid*, *cycle*. The task is to determine whether or not the node is a part of the motif. For the graph classification task, each graph consists of a base graph randomly attached by one type of motif. The task is to classify what type of motifs the graph contains.

We also test D4Explainer over real-world datasets, Cornell, Mutag, BBBP, and NCI1. Mutag, BBBP and NCI1 are molecular datasets where each graph is labeled as either having a specific chemical property or not. For Mutag, the mutagenicity of a molecule is linked to the presence of electron-attracting elements combined with nitro groups (such as NO2). Additionally, molecules containing three or more fused rings are more likely to be mutagenic compared to those with one or two rings [64]. Cornell is a webpage dataset introduced by [52]. Nodes are web pages, and edges are hyperlinks between them. Node features are bag-of-words representations of web pages. Nodes are classified into one of five categories: Students, Projects, Courses, Faculty, and Staff. Cornell is a highly heterophilous dataset, *i.e.,* the adjacent nodes tend to have different features and labels, which further poses a challenge to the explanation task. Nonetheless, there is no explicit motif that leads to a specific class in real-world datasets. The statistical information of all datasets is summarized in Table 6. We use different types of target GNNs to evaluate the performance of D4Explainer , including GCN, GIN, and EGNN. The last row shows the test accuracy of the target GNN. Each target GNN achieves more than $80\%$ accuracy over the test dataset.

## E.2    Metrics

We use the following metrics to evaluate the generated explanations, where the modification rate (MR) is our proposed adjustment to the sparsity metric used in previous works [13, 36].

- **Counterfactual Accuracy (CF-ACC)** [13] measures whether the explainer can generate effective counterfactual explanations. It is formulated as the proportion of generated explanations that change the model's prediction.

$$\text{CF-ACC} = 1 - \frac{1}{|\mathcal{G}|} \sum_{G^o \in \mathcal{G}} \left( \mathbb{1}(\hat{Y}_{G^c} = \hat{Y}_{G^o}), \right. \tag{13}$$

  where $G^o$ is the original graph, $G^c$ is the generated counterfactual explanation regarding $G^o$ , $\mathcal{G}$ is the test dataset and $|\mathcal{G}|$ denotes the size of $\mathcal{G}$. $\hat{Y}_G$ is the label of $G$ predicted by the target GNN $f$. $\mathbb{1}(\cdot)$ is the indicator function to check whether $\hat{Y}_{G^c}$ equals to $\hat{Y}_{G^o}$. Since we aim to generate counterfactual explanations, a higher CF-ACC is better.

- **Fidelity** [3, 36] quantifies the change in predicted probability over the original class. It is formulated as

$$\text{Fidelity} = \frac{1}{|\mathcal{G}|} \sum_{G^o \in \mathcal{G}} \left[ f(G^o)[\hat{Y}_{G^o}] - f(G^c)[\hat{Y}_{G^o}] \right], \tag{14}$$

  where $f(G)[\hat{Y}]$ denotes the probability output of the model $f$ for graph $G$ over class $\hat{Y}$. A higher fidelity score indicates better counterfactual explanations.

- **Modification ratio** is the proportion of changed edges:

$$\mathrm{MR} = \frac{(\text{\# of deleted edges} + \text{\# of added edges})}{|E|}, \tag{15}$$

where $|E|$ is the number of edges in the original graph. For baseline models that only consider deleting edges, MR can be easily adjusted to the proportion of deleted edges with respect to the original graph, which is the sparsity metric used in prior works [13, 36].

## E.3 Model Parameters

Table 7: Optimal parameters for each dataset

|            | num hidden | num layers in PPGN | batch size | alpha |
| --- | --- | --- | --- | --- |
| BA-shapes  | 64  | 6 | 4  | 0.005 |
| Tree-Cycle | 64  | 6 | 32 | 0.1   |
| Tree-Grids | 128 | 8 | 32 | 0.05  |
| Cornell    | 128 | 6 | 4  | 0.05  |
| BA-3Motif  | 128 | 6 | 32 | 0.05  |
| Mutag      | 64  | 6 | 2  | 0.001 |
| BBBP       | 128 | 6 | 16 | 0.005 |
| NCI1       | 128 | 6 | 32 | 0.01  |

In the implementation, we need to perform multi-step diffusion. We uniformly sample $\bar{\beta}_t$ in the range of $[0, 0.5]$ to control the level of noise and generate the graph $G_t$ with the corresponding level of noise. Given each $G_t$, the denoising model is trained to recover the clean graph $\tilde{G}_0$. During the training stage, we employ Adam [65] as our optimizer and ExponentialLR [66] as the scheduler. Table 7 shows the optimal numbers of hidden units, layers in PPGN, batch size, and the regularization coefficient $\alpha$ for each dataset. We run 1500 epochs and set the initial learning rate as $1 \times 10^{-3}$ across all datasets.

## E.4 In-distribution Evaluation

MMD (Maximum Mean Discrepancy) is a metric used to compare the distance between two probability distributions. In the context of graph statistics, MMD can be used to compare the degree distribution, cluster coefficient distribution, and spectrum distribution. MMD is also widely used for accessing the distribution-learning ability of graph generative models [61, 19, 20, 62, 44].

A graph's degree distribution represents the frequency of nodes with different degree values in the graph. The clustering coefficient of a node is a measure of the node's local clustering or the fraction of triangles that the node participates in. Spectrum distribution refers to the distribution of eigenvalues of the adjacency matrix or Laplacian matrix of a graph, which can be used to study the graph's structure and dynamics. The MMD between two sets of samples from distributions $p$ and $q$ can be formulated as

$$\mathrm{MMD}^2(p\|q) = \mathbb{E}_{x,y\sim p}[k(x,y)] + \mathbb{E}_{x,y\sim q}[k(x,y)] - 2\mathbb{E}_{x\sim p,y\sim q}[k(x,y)], \tag{16}$$

where $k(x, y)$ denotes the kernel function. Following the in-distribution evaluation setting in [61, 19, 20, 62, 44], we use Gaussian Earth Mover's Distance kernel to compute the MMDs of degree distributions, clustering coefficients, and spectrum distributions. Complete MMD results are shown in Table 8.

## E.5 Diversity Evaluation

Figure 8 shows the generated counterfactual examples for BA-shapes, Tree-Cycle, Tree-Grid, and BA-3Motif. The graphs in the first row are the original graphs to explain. The second row shows the counterfactual examples generated by CF-GNNExplainer. The last two rows show two types of counterfactual examples generated by D4Explainer . Labels at the bottom right indicate the motif contained in the graph. We find that the easiest way to generate counterfactual explanations is to destroy the original motif by deleting essential edges, which we call "corruption".

Table 8: MMD distances between the generated explanations and test graphs. We report MMD distances of degree distributions (*Deg.*), cluster coefficients (*Clus.*), and spectrum distributions (*Spec.*). *Sum.* means the summation of the previous three metrics. We bold the smallest value and underlined the second-best value in each column.

| Models | BA-3Motif | | | | Mutag | | | | BBBP | | | | NCI1 | | | |
|---|---|---|---|---|---|---|---|---|---|---|---|---|---|---|---|---|
| | Deg. | Clus. | Spec. | Sum. | Deg. | Clus. | Spec. | Sum. | Deg. | Clus. | Spec. | Sum. | Deg. | Clus. | Spec. | Sum. |
| RamdomCaster | 0.2336 | 0.0574 | 0.0532 | 0.3442 | 0.1593 | 0.0247 | 0.0417 | 0.2257 | 0.1693 | 0.0072 | 0.0397 | 0.2162 | 0.1847 | 1.9769 | 0.0404 | 2.2020 |
| GNNExplainer | 0.2366 | 0.0803 | 0.0531 | 0.3700 | 0.1614 | 0.0002 | 0.0409 | 0.2025 | 0.1615 | 0.0002 | 0.0395 | 0.2012 | 0.1577 | 0.0005 | 0.0405 | 0.1987 |
| SAExplainer | 0.2431 | **0.0108** | 0.0534 | 0.3073 | **0.0940** | 0.0032 | 0.0412 | **0.1384** | 0.1594 | 0.0032 | 0.0402 | 0.2028 | 0.189 | 0.0002 | 0.0408 | 0.2300 |
| GradCam | 0.2224 | 0.0825 | 0.0539 | 0.3588 | 0.1122 | 0.0083 | 0.0416 | 0.1621 | 0.0699 | 0.0026 | 0.0384 | 0.1109 | 0.1638 | 0.0003 | 0.0404 | 0.2045 |
| IGExplainer | 0.2474 | 0.0436 | 0.0533 | 0.3443 | 0.1292 | **0.0000** | 0.0411 | 0.1703 | 0.0908 | **0.0000** | 0.0394 | 0.1302 | 0.4288 | 0.0002 | 0.0398 | 0.4688 |
| PGExplainer | 0.2459 | 0.0308 | 0.0628 | 0.3395 | 0.1475 | 0.0002 | 0.0418 | 0.1895 | 0.2014 | 0.0018 | 0.0403 | 0.2435 | 0.1937 | **0.0000** | 0.0396 | 0.2333 |
| PGMExplainer | 0.2493 | 0.0246 | 0.0543 | 0.3282 | 0.1800 | 0.0002 | 0.0419 | 0.2221 | 0.1916 | 0.0003 | 0.0403 | 0.2322 | 0.2199 | **0.0000** | 0.0404 | 0.2603 |
| CXPlain | 0.2356 | 0.0412 | 0.0535 | 0.3303 | 0.1734 | 1.2706 | 0.0417 | 1.4857 | 0.1768 | 0.0001 | 0.0394 | 0.2163 | 0.1629 | 0.0001 | 0.0404 | 0.2034 |
| CF-GNNExplainer | 0.1669 | 0.0366 | 0.0531 | 0.2566 | 0.1172 | **0.0000** | 0.0380 | 0.1552 | 0.0870 | 0.0001 | 0.0393 | 0.1264 | 0.1224 | 0.0001 | 0.0404 | 0.1629 |
| **D4Explainer(ours)** | **0.1028** | 0.0265 | **0.0517** | **0.1810** | 0.1172 | **0.0000** | **0.0244** | 0.1416 | **0.0530** | **0.0000** | **0.0331** | **0.0861** | **0.1006** | **0.0000** | **0.0353** | **0.1359** |

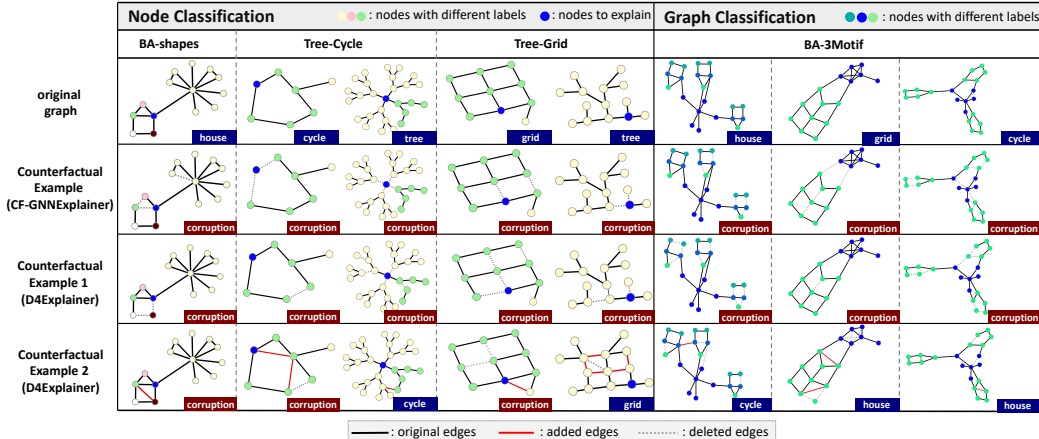

Figure 8: Counterfactual explanations generated by CF-GNNExplainer and D4Explainer over four datasets. The labels at the bottom right represent the type of generated counterfactual explanations. CF-GNNExplaienr can only corrupt the original motif, while D4Explainer can corrupt the original motif and create the counterfactual motif.

**Analysis.** We observe from Figure 8 that the previous explainer alters the model's prediction by corrupting the original motif as its counterfactual explanation. Moreover, in the explanation task for node classification, previous explainers tend to remove the connection between the node to be explained and its neighbors. For example, in the second row of BA-shapes, Tree-Cycle, and Tree-Grid, CF-GNN-Explainer deletes the edges between the blue node and its neighboring nodes. One possible reason is that the isolated node cannot receive the messages from its neighbors with the well-trained GNN, thus naturally becoming an out-of-distribution sample and degrading the model's prediction confidence. Instead, D4Explainer generates counterfactual explanations not only by destroying the original motif but also by creating the truly counterfactual motifs, as shown in the last row of Figure 8. It is noticeable that D4Explainer can identify the counterfactual motif and complete that based on the original graph, which is hardly achieved by previous methods that only consider the edge deletion.

## E.6 Complexity and Inference Time Evaluation

The computation of the loss function involves four steps. (1) uniformly sample $t$ from $[0, T]$, (2) apply forward diffusion process $q(G_t|G_0)$, (3) calculate $p_\theta(G_0|G_t)$ via the denoising network and (4) sample counterfactual graphs from $p_\theta(G_0|G_t)$. Let $N$ denote the number of nodes in the original graph. The time complexity is $\mathcal{O}(1)$ for Step(1) and $\mathcal{O}(N^2)$ for Step(2) and $\mathcal{O}(N^3)$ for Step(3) due to the matrix multiplication. Step(4) results in a time complexity of $\mathcal{O}(N^2)$ for sampling. Overall, the time complexity of D4Explainer is mainly determined by the denoising network.

To empirically evaluate the efficiency of D4Explaienr, we conduct the runtime comparison between D4Explainer and baselines. The results are shown in Table 9. Except for PGExplainer, other baselines reported in Table 9 are non-generative, that is, the model optimizes an explanation for input instances

one by one during the inference stage. Therefore, these models require more time to generate one explanation and become less efficient. On the contrary, D4Explainer incorporates the generative graph distribution learning into the optimization objective and captures the underlying distribution of the explanation graphs over the entire dataset. Consequently, D4Explainer is relatively efficient during the inference stage.

Table 9: Runtime analysis of all baselines. We compute mean/std values of inference time to generate explanations for a single instance

|  | GNNExplainer | IGExplainer | PGExplainer | PGMExplainer | CXPlain | CF-GNNExplainer | D4Explainer |
|---|---|---|---|---|---|---|---|
| **Tree-Cycle** | 1.367±0.023 | 2.684±0.368 | 0.028±0.007 | 1.145±0.012 | 1.427±0.277 | 2.637±0.540 | 0.022±0.002 |
| **Mutag** | 1.492±0.037 | 3.157±0.454 | 0.035±0.005 | 1.576±0.038 | 1.842±0.320 | 2.741±0.536 | 0.030±0.006 |

### E.7    Robustness Evaluation

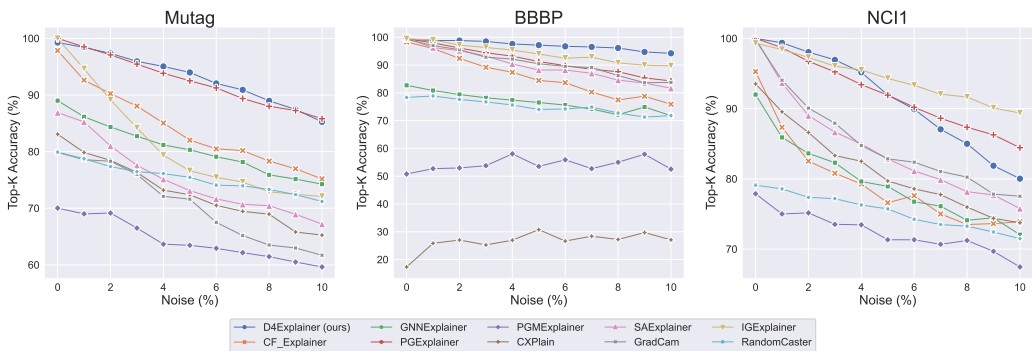

Figure 9: Robustness performance of all methods. The modification ratio is controlled as $20\%$ across all methods.

Following previous setup [36], we identify the $K$ most relevant edges in the original counterfactual explanation and compute the fraction of these edges present in the explanation of its noisy version, denoted by Top-$K$ Accuracy. We apply noise by randomly adding or removing edges with probability $\sigma$. We restrict that $\sigma < 0.1$, as the noise of larger $\sigma$ may cause the noisy graph to switch the predicted label. Top-$K$ accuracy *w.r.t.* noise levels over three molecular datasets are shown in Fig. 9. As shown in Figure 9, we observe that D4Explainer outperforms all baselines on BBBP and performs comparably to PGExplainer on Mutag and IGExplainer on NCI1. To keep consistent, we show Top-$K$ Accuracy with noise levels from $0\%$ to $10\%$ for three datasets. However, with more than $5\%$ noise, only $65\%$ of perturbed noisy graphs have the same label as the original one for NCI1, much smaller than BBBP and Mutag. The high sensitivity of NCI1 to noise explains the drop in robustness as noise increases past $\sigma = 0.05$. Overall, results in Figure 9 demonstrate D4Explainer's strong ability to generate consistently effective counterfactual explanations despite the noise.

### E.8    Model-level Explanation

Table 10 shows the quantitative comparison of probability and density in the model-level explanations generated by XGNN and D4Exlainer. We generate 100 model-level explanations for each dataset and compute the average probability (*i.e.,* explanation confidence) and average density. As can be observed from Table 10, D4Explainer outperforms XGNN over both metrics on four datasets in general. Particularly, D4Explainer achieves almost $100\%$ explanation confidence on three synthetic datasets with an appropriate $N$, *i.e.,* the number of nodes in the desired explanation. In many real-world scenarios, the ground truth model-level explanations are not unique. That is, we can hardly know the exact discriminative graph structure and feature that the GNNs learned for prediction. The appropriate $N$ might require domain-specific knowledge, while we can test with different $N$ and select the one that achieves the highest explanation confidence.

Table 11 reports the sensitivity of $K$ and $T$, which denote the number of candidates and number of iterations in the reverse sampling algorithm for model-level explanations, respectively. Similarly, we utilize probability and density to quantitatively measure the properties of the generated explanations.

Table 10: Quantitative comparison in terms of probability and density with different numbers of nodes

| | | BA-3Motif | | | Mutag | | | Tree-Grid | | | Tree-Cycle | | |
|---|---|---|---|---|---|---|---|---|---|---|---|---|---|
| | # nodes | 5 | 6 | 7 | 6 | 7 | 8 | 8 | 9 | 10 | 5 | 6 | 7 |
| D4Explainer | Prob. | **0.997** | **1.000** | **0.998** | **0.832** | **0.856** | **0.920** | **0.832** | **0.994** | **0.991** | **0.991** | **0.995** | 0.989 |
| | Density | **0.313** | **0.327** | **0.294** | **0.278** | **0.327** | **0.315** | **0.369** | **0.372** | **0.379** | 0.400 | **0.381** | **0.343** |
| XGNN | Prob. | 0.632 | 0.883 | 0.834 | 0.523 | 0.824 | 0.875 | 0.752 | 0.836 | 0.902 | 0.968 | 0.989 | **0.992** |
| | Density | 0.552 | 0.444 | 0.433 | 0.537 | 0.479 | 0.437 | 0.421 | 0.406 | 0.439 | 0.400 | 0.390 | 0.367 |

Table 11: Hyperparameter sensitivity in model-level explanation generation

| hyper-parameters | BA-3Motif (N=5) | | Mutag (N=6) | | Tree-Grid (N=9) | | Tree-Cycle (N=6) | |
|---|---|---|---|---|---|---|---|---|
| | Prob. | Density | Prob. | Density | Prob. | Density | Prob. | Density |
| (1) $K = 10, T = 50$ | 0.899 | 0.3152 | 0.799 | 0.314 | 0.901 | 0.400 | 0.987 | 0.372 |
| (2) $K = 20, T = 10$ | 0.798 | 0.3277 | 0.524 | 0.284 | 0.897 | 0.383 | 0.991 | 0.388 |
| (3) $K = 20, T = 50$ | 0.967 | 0.3126 | 0.812 | 0.295 | **0.994** | 0.372 | 0.994 | 0.361 |
| (4) $K = 20, T = 100$ | **0.997** | 0.3133 | **0.832** | **0.278** | 0.972 | 0.427 | 0.992 | **0.325** |
| (5) $K = 30, T = 50$ | 0.972 | **0.2972** | 0.823 | 0.287 | **0.994** | **0.355** | **0.997** | 0.361 |

From the table, we can observe that when $T$ is small (*e.g.*, $T = 10$), the probability $p$ is relatively lower (see experiments 2, 3, 4), indicating that the quality of the generated model-level explanations is sub-optimal. This result further emphasizes the effectiveness of multi-step sampling, which progressively increases the explanation confidence. Additionally, we observe that the probability $p$ increases as $K$ increases, under the same conditions of $T$ (see experiments 1, 3, 5). This suggests that increasing the number of candidates helps to obtain a good-quality model-level explanation within fewer steps. In our implementation, we ensure $K \geqslant 20$ and $T \geqslant 50$ by default for a balance between the quality and time complexity.

## F    Discussions

**Limitations**. In this paper, we explore the application of denoising diffusion models in generating counterfactual and model-level explanations for Graph Neural Networks (GNNs). While D4Explainer has shown promising results in terms of various metrics, including explanation accuracy, robustness, diversity, *etc.*It still introduces unique challenges and limitations. Firstly, the computational complexity of training D4Explainer on large-scale graph structures poses scalability concerns. Additionally, the reliance on the underlying GNN architecture can limit the generalizability of the explanations across different GNN models. Furthermore, in model-level explanations, high-quality explanations rely on an appropriate number of nodes, which might require domain-specific knowledge.

**Broader Impacts**. The social impact of this work is twofold. On one hand, the ability to generate counterfactual explanations for GNNs can enhance transparency and interpretability, empowering users to understand and trust the decisions made by these models. By shedding light on the features and interactions that contribute to specific predictions, this work can facilitate the identification of biases, discriminatory patterns, and vulnerabilities present in GNNs. However, it is crucial to acknowledge the limitations and potential risks associated with using generated explanations as they might not always capture the complete causal relationships present in complex graph structures. Consequently, relying solely on these explanations may lead to unintended consequences, such as reinforcing existing biases or making incorrect assumptions about the model's behavior.

**Future Works**. Moving forward, several important avenues for future research emerge from this study. First, addressing the scalability challenges associated with training denoising diffusion models on large-scale graph structures is a crucial direction. Developing efficient training algorithms, exploring parallelization strategies, and investigating graph-specific optimizations can significantly improve the applicability of D4Explainer to real-world large-scale graphs. Secondly, an interesting future direction is to consider the node attributes and edge attributes during the explanation generation, *e.g.*, by performing diffusion processes over continuous features. Moreover, future work should address the potential risks associated with unintended consequences, biases, and misuse of explanations. Developing guidelines and frameworks for responsible and accountable use of generated explanations is crucial, particularly in high-stakes domains such as healthcare, finance, and criminal justice.

