# OpenReview forum: "D4Explainer: In-distribution Explanations of Graph Neural Network via Discrete Denoising Diffusion"
_NeurIPS.cc/2023/Conference — NeurIPS 2023 poster_

### Official Review · Reviewer_ofrf · 2023-07-01

**Soundness:** 3 good
**Presentation:** 2 fair
**Contribution:** 2 fair
**Rating:** 6
**Confidence:** 4

**Summary:**

D4Explainer contributes to the important problem of explainable Graph Neural Networks (GNNs) and offers in-distribution GNN explanations for both counterfactual and model-level scenarios. The incorporation of generative graph distribution learning into the optimization objective allows D4Explainer to generate diverse counterfactual graphs and identify discriminative graph patterns, enhancing the reliability and informativeness of the explanations. The empirical evaluation results are promising. Clarifications are needed to help readers better understand D4Explainer.

I have read the author’s rebuttal and raised the overall rating.

**Strengths:**

D4Explainer integrates graph diffusion into producing counterfactual and model-level explanations. While individual components have been widely investigated, their combination in making GNN more explainable is new.
The experimental results are good.
The manuscript is in general well organized.

**Weaknesses:**

Some parts of the manuscript need clarifications (see Questions).
The experiments were only conducted on small graphs (<=10 nodes).
Edges are constrained to have categorical properties. In many real applications, they are associated with continuous attributes.
Node addition/deletion is considered.
It is unclear how much effect N (number of nodes in graph) has on experimental results. How to choose a right N?
The “Explanation Diversity Evaluation” only offers a few examples, not even at the qualitative level. It is not clear how generalizable those observations are.

**Questions:**

1. Figure 1 is confusing. In the top plot, if each dot is a “node to explain”, why the blowouts of the counterfactual examples are graphs? Does the distances between a “node” and its counterfactual examples indicate anything?
2. Eq (3) seems computationally expensive. The complexity analysis in Section 4.3 doesn't seem to provide a full scale analysis.
3. The derivation of Eq (4) needs explanations.
4. The model-level explanation “aims to identify recurring and discriminant graph patterns that can trigger a specific prediction from the model f”. It is not clear if the calculation of G^m ensures “recurring”. Under what circumstance, this is true with high probability.
5. Since “proximity to the original graph” is used in deriving the proposed approach, why not report an index based on “proximity to the original graph” in the experimental results, instead of MMD?
6. "We report AUC values computed over 10 modification ratios from 0 to 0.3". What does this mean: Several modification ratios were tried, and the best result is reported?

**Limitations:**

This is a rather basic research. No potential negative societal impact is detected.

---

> ### Author Rebuttal · Authors · 2023-08-10
>
> We sincerely appreciate your valuable feedback, which significantly improves the quality of this work. We would like to answer the questions you raised as follows.
>
> > The experiments were only conducted on small graphs (<=10 nodes)
>
> We prudently clarify that the above comment is incorrect. Please refer to Table 6 in Appendix E.1 for dataset information. For the node classification task, the graph datasets we used in experiments contain hundreds of nodes and thousands of edges. For graph classification tasks, we are using molecular datasets with around 30 nodes. Our choice of datasets is in line with the prevailing standard in the field of GNN explainability.
>
> >Edges are constrained to have categorical properties
>
> The diffusion process and denoising network can indeed be applied to continuous/categorical node/edge attributes. We've discussed these aspects in detail in the main text (including Appendix C.1 for discrete diffusion and C.2 for continuous attributes). We believe it would be an interesting future extension to consider node/edge attributes.
>
> >How much effect N has on experimental results? How to choose the right N?
>
> Good question! In the model-level explanation task, we conduct a hyper-parameter sensitivity analysis on $N$ in Table 10 in Appendix E.8. We observe that model-level explanations with different $N$ can achieve comparably high confidence scores, which indicates that D4Explainer can find the best graph structures, given any $N$. We have to admit that choosing the right/best $N$ requires domain-specific knowledge. In many real-world scenarios, the ground truth model-level explanations are not unique. A simple and feasible method is testing with different numbers of nodes $N$ and selecting the ones that achieve high confidence scores.
>
> > The “explanation diversity evaluation” only offers a few examples. How generalizable?
>
> Complete results of “Explanation Diversity Evaluation” are given in Figure 8 in Appendix E.5. Diversity is generalizable for different labels on four synthetic datasets. Here, we would like to emphasize the core insight is that D4Explainer is able to detect the distinctive motifs and truly convert the original motifs to counterfactual ones, instead of simply destroying the original motifs.
>
> >Clarification to Figure 1
>
> In Figure 1, each point represents the t-sne projection of the embedding of a node to explain, which is computed over its computational graph. Counterfactual examples generated by explainers are essentially the modified computational graph of the node to explain, which leads to the different prediction labels on the central node. The embedding distance between two nodes reflects how similar or proximate with each other.  The embedding of a valid counterfactual explanation should be close to the average embedding of all instances across the dataset. In Figure1, the embedding of the counterfactual example generated by CF-GNNExplainer falls far from the average embedding of in-distribution instances with the Cycle label. We believe it is a result of the OOD effect, making the explanations unreliable.
>
> >What is the complexity of Eq (3) and full complexity analysis?
>
> Thanks for the question! The time complexity of Eq(3) involves (1)uniformly sample t from [0,T] (2) applying forward diffusion process $q(G_t|G_0)$ and (3) calculate $p_{\theta}(G_0|G_t)$ via the denoising network and (4) sample counterfactual graphs from $p_{\theta}(G_0|G_t)$. Let $N$ denote the number of nodes in the original graph. The time complexity is $O(1)$ for Step1 and $O(N^2)$ for step 2. For step3, we analyze the complexity in Sec 4.3. Step 4 results in a time complexity of $O(N^2)$ for sampling. Overall, the time complexity of Eq(3) is mainly determined by the denoising network. We have revised Sec 4.3 to involve a complete discussion on computational complexity.
>
> >Explanations to the derivation of Eq (4)
>
> Thanks for bringing this up! To provide a high-level understanding of Eq(4), the generation path from $G_t^r$ to $G_{t-1}^r$ is actually $G_t^r \rightarrow G_0 \rightarrow G_{t-1}^r$, where the first step is controlled by $p_{\theta}( G_0 |G_{t}^r)$. Second step is controlled by $q(G_{t-1}^r|G_0)$. $f(C|G_0)$ is multiplied to condition the reverse sampling on the target C. We will provide complete derivation and proof in Appendix in our revised version.
>
> >How does the calculation of G^m ensure “recurring” property?
>
> The calculation of $G^m=argmax_G P_f(C_i|G)$ is first proposed in the previous work [1] and then widely used in the problem formulation. We agree that the “recurring” property can hardly be guaranteed by the above calculation alone. In D4Explainer framework, the “recurring” property is actually ensured by incorporating graph distribution learning into the optimization objective. In addition, we employ a candidate selection strategy to ensure the generated model-level explanation maximizes the likelihood of $P_f(C_i|G_m)$. We will clarify more about how we ensure “recurring” property in Sec.4.2 in our revised version.
>
> [1]XGNN: Towards Model-Level Explanations of Graph Neural Networks
>
> >Clarification to “We report AUC values computed over 10 modification ratios from 0 to 0.3”
>
> Since the accuracy of counterfactual explanations is sensitive to the modification ratio, we report the accuracy curve over 10 different modification ratios from 0 to 0.3 (See Figure 4). We denote the area under this curve as CF-ACC AUC and report the results in Table 1.
>
> >Why not report an index based on “proximity to the original graph”, Instead of MMD?
>
> “Proximity” actually can be represented by the “Modification Ratio”, which has already been adopted as a metric in our experiments. Compared with “Proximity”, MMD is a global metric, which measures the distribution distance between two sets of graph samples, instead of two graphs.

---

> > ### Comment · Reviewer_ofrf · 2023-08-15
> >
> > Thank authors for providing more information. I raised by overall score.

---

### Official Review · Reviewer_Tz8g · 2023-07-05

**Soundness:** 2 fair
**Presentation:** 1 poor
**Contribution:** 2 fair
**Rating:** 5
**Confidence:** 5

**Summary:**

The authors propose to apply discrete denoising diffusion as a technique for generating counterfactual and model-level graph explanations. From my understanding, the proposed method focuses solely on incorporating graph structure while **disregarding node features**. Additionally, the authors claim that the proposed framework can generate in-distribution graphs, unlike other methods. However, their evaluating method primarily involves comparing graph statistics, which may not necessarily capture the true essence of graph distribution.

**Strengths:**

- The main strength of the paper is applying the diffusion model to generate both counterfactual and model-level explanations.

- Contemplating the possibility of the subgraph being ood is interesting. However, without supporting evidence, it remains a hypothesis without validation in this context.

**Weaknesses:**

- The paper has the tendency to include a few unsupported and wrong claims. A few examples are provided here:

1) [D4Explainer explores counterfactual explanations in a larger search space than prior works by allowing adding edges, which provides a new perspective of generating counterfactual explanation] -> This is not true as a few of the previous works already are based on other kinds of generative models, for example, CLEAR (NeurIPS 2022).

2) The authors referred to the CLEAR paper (reference # 37) [only edge deletion is allowed as in the previous paradigm [36, 13, 37]] which is a wrong claim.

3) There is no theoretical or empirical proof that the generated graphs are in-distribution. [However, generating in-distribution graphs is challenging, due to the difficulty of encoding complex graph distributions, e.g., the distribution of node degrees, cycle counts and edge homogeneity.] As far as I can say, there is a difference between graph distribution and the distribution of graph statistics such as node degrees.  Empirical evidence or a citation is needed before being on top of this assumption.

- The explainability model cannot handle node attributes and only focus on graph structure.

- The other weakness of the paper is related works. The authors not only need to include previous works such as CLEAR as a baseline but also need to include them in their related works discussing the difference between the proposed framework and the existing ones.

- Evaluation is not satisfying as it does not include std. Apart from that, the authors did not more informative metrics to compare the models including Validity, Proximity, as well as causality which are important for counterfactual explainability.

**Questions:**

1) Figure 1 does not make sense to me. Why t-sne projections have been used for graph structure data and how this is related to ood is not clear to me. The misclassification might not necessarily be related to the ood effect. An empirical study or a theoretical proof is needed?

2) What is the guarantee for the diffusion base generator, to generate in-distribution graphs? Citation or empirical results are needed. **Note:** Even when considering the MDD statistics, which I believe is an inadequate metric for evaluating this, the generated graphs may not be in-distribution depending on the dataset, e.g. Mutag.

3) Number of modifications is not a useful metric as does not consider the size of the graph. Why not consider sparsity?

4) Why not AUC? What about the ACC for the simulated graphs? Please also include validity and proximity metrics.

5) all results need std as well.

6) MDD table does not support the claim of the paper and it is not clear in which dataset the model works better.

7) Model-level explanation experiments are not included that many datasets and in their current shape mostly can be seen as an ablation study. More baselines are needed for this part as well.

8) It is not clear to me how the input is perturbed for robustness evaluation.

9) It is not clear to me how the method can conditionally generate counterfactual explainability for the property of the interest.

**Limitations:**

The authors mention some of the limitation of the proposed method.

---

> ### Author Rebuttal · Authors · 2023-08-10
>
> We sincerely appreciate your valuable time and efforts in reviewing this paper. We would like to answer potential concerns as follows.
> >Previous generative works, eg. CLEAR. "[Only edge deletion is allowed as in the previous paradigm[36,13,17] is a wrong claim.”
>
> We apologize for the misstatement here. Yes, CLEAR allows edge addition and we actually obtained many insights from it. We will revise the statements as follows. “Previous counterfactual explainers typically only consider edge deletion, leaving the effect of edge addition underexplored.” Here we would like to emphasize the core innovation of our D4Explainer.
> - The denoising strategy enhances the robustness of generated explanations, which is not studied in CLEAR.
> - Our model is the first to explore both instance-level and model-level explanations within a unified framework, while CLEAR focuses on counterfactual explanations only.
> - We included high-level understandings of how edge addition helps to create truly counterfactual motifs in Sec.5.2.2 which is not discussed in detail in CLEAR.
>
> >No theoretical/empirical proof for in-distribution graph.
>
> We note our investigations focus on assessing how discrete graph structures affect explainability properties. To support this claim, we have conducted extensive in-distribution evaluations in Sec 5.2.1. Previous studies[1] have shown that MMD is a versatile and necessary metric for comparing graph distribution. MMD is also widely used for accessing the distribution-learning ability of graph generative models in this field [2,3,4]. We believe our empirical results in Sec 5.2.1 can sufficiently demonstrate the in-distribution property in terms of graph structures.
> >The model cannot handle node attributes.
>
> The diffusion process and denoising network can indeed be applied to continuous/categorical node/edge attributes. We've discussed these aspects in detail in Appendix C.1 for discrete diffusion and C.2 for continuous attributes. We believe it would be an interesting future extension to consider node/edge attributes.
>
> >Include CLEAR as a baseline
>
> We tried to involve CLEAR as a baseline but we didn’t find any publicly available codes for CLEAR before the submission deadline. Now we reproduce CLEAR based on the original paper and report preliminary results as follows (complete results in our revised version). We hope that the experimental results could improve your confidence in our work.
> |             | Tree-Cycle  | BBBP        |
> |-------------|-------------|-------------|
> | D4Explainer | 0.917±0.012 | 0.781±0.024 |
> | CLEAR       | 0.834±0.010 | 0.754±0.035 |
>
> | BBBP        | Deg.   | Clus.  | Spec.  | Sum.   |
> |-------------|--------|--------|--------|--------|
> | D4Explainer | 0.0530 | 0.0000 | 0.0331 | 0.0861 |
> | CLEAR       | 0.0782 | 0.0000 | 0.0329 | 0.1109 |
> >Evaluation does not include std.
>
> We will include std results in Appendix in our revised version.
>
> >Why t-sne and how this is related to ood issue?
>
> In Figure 1, each point represents the t-sne projection of the embedding of a node to explain. The embedding of a valid counterfactual example should be close to the average embedding of all instances across the dataset. However, we observe that the embedding of the example generated by CF-GNNExplainer falls far from the distribution, which raises suspicions of being OOD.
> >What is the guarantee for in-distribution graphs?
>
> The key guarantee lies in their training process. The model learns to reverse the diffusion process and denoise the initially added noise. This is done by using maximum the likelihood of generating the original graph. Many previous studies [2,3,4] demonstrate the distribution-learning ability of diffusion-based generators.
> >Number of modifications does not consider the graph size. Why not sparsity?
>
> We actually use the Modification ratio (MR), which considers the size of the original graphs. Sparsity is usually used in factual explainability. In counterfactual explanation tasks, we emphasize the modification is minimal based on the given graph.
> >Why not AUC? What about ACC, validity, proximity and causality metrics?
>
> We have reported both AUC and ACC metrics in the paper. We report the accuracy curve over 10 different MRs (in Figure 4). We report the area under this curve as CF-ACC AUC in Table 1. We believe validity is the same as the accuracy and proximity w.r.t. graph structure is similar to the MR in our submission. The only difference is that we normalize it by the number of original edges. Regarding casualty, we think it is not a widely used metric in counterfactual explanations, since it highly depends on domain knowledge and dataset description.
> >Model-level experiments do not include that many datasets and baselines.
>
> The datasets for model-level explanations actually include cycle, ring, grid, house motifs, and real-world datasets (e.g. mutag), which cover almost all motifs tested in previous works. We also tried to involve GNNInterpreter as a baseline, but there are no publicly available codes yet.
> >How is the input perturbed?
>
> For robustness evaluation, we randomly add new edges or delete original edges with probability $\sigma$ over any pair of nodes in the original graph.
> > How can the method conditionally generate counterfactual explainability?
>
> In the counterfactual explanation task, we design $L_{cf}$ to ensure the counterfactual property, which minimizes the likelihood of the generated explanation being predicted as the original label. $L_{cf}$ can also be easily extended to ensure other target properties of interest by replacing  $\hat{Y}_{G_0}$.
>
> [1]Evaluation Metrics for Graph Generative Models: Problems, Pitfalls, and Practical Solutions
>
> [2]DiGress: Discrete Denoising diffusion for graph generation
>
> [3]Diffusion Models for Graphs Benefit From Discrete State Spaces
>
> [4]GraphGUIDE: interpretable and controllable conditional graph generation with discrete Bernoulli diffusion

---

> > ### Comment · Reviewer_Tz8g · 2023-08-17
> >
> > Thanks to the reviewer to address some of my main concerns.
> >
> > - I still believe that the authors should be careful about their claims.
> >
> > - Add new experiments which compare the performance with CLEAR as STOA baselines.
> >
> > - Consider metrics similar to CLEAR for evaluation.
> >
> > - And make it clear how the model can generate counterfactual explainability, conditional. I didn't get my answer in this regard.
> >
> > For now, I increased my score to 4!

---

> > > ### Author Response · Authors · 2023-08-20
> > > **Response to Reviewer Tz8g**
> > >
> > > Thanks for replying and increasing the score. We would like to answer your questions and address potential concerns as follows.
> > >
> > > 1. We totally agree with your comments. We will take extra care in articulating the novelty and distinct contributions of our work, particularly in relation to the impact of edge addition and the broader advantages of our proposed framework. (e.g., robustness, in-distribution property w.r.t. graph structure, and a unified framework encompassing both instance-level and model-level explanations).
> > > 2. In response to your suggestion, we have incorporated CLEAR as one of our baselines in our experiments. We will report the complete results (as well as std.) in our final version.
> > > 3. To clarify the evaluation metrics, we would like to emphasize that we have already incorporated metrics akin to CLEAR's validity and proximity$_A$ into our experimentation. Specifically, in our work, validity aligns with CF-ACC, and proximity$_A$ bears a resemblance to the modification ratio. The difference lies in our normalization approach, wherein we normalize by the number of original edges. Regarding Causality, an essential aspect of CLEAR, we acknowledge its dependence on dataset-specific causal relations and constraints. Given the ambiguity in real-world datasets' ground-truth causality, we believe evaluating Causality metrics could be challenging, necessitating profound knowledge specific to the dataset.
> > > 4. The proposed D4Explainer is capable of conditional generating counterfactual explanations. Given a desired predicted label $Y^\star$, we can rewrite the counterfactual loss as  $\mathcal{L}_{cf}=-\mathbb{E}$$ _{q(G_0)}\mathbb{E}$$ _{t \sim[0, T]}\mathbb{E}$$ _{q(G_t \mid G_0)} \mathbb{E}$$ _{p_θ}$$ _{(\tilde{G}_0 \mid G_t)}$$ \log (f(\tilde{G}_0)[Y^\star])$. In this way, D4Explainer is optimized by maximizing the likelihood of the counterfactual explanation conditioned on the target label  $Y^\star$.
> > > For consistency with existing literature [1], we formulate the counterfactual explanation task as generating examples with different labels from the previous ones, without assigning a specific label in this work.
> > >
> > > Thanks again for your engagement. We will definitely adopt your suggestions to further improve paper quality and writing. Let us know if you have any further questions/concerns.
> > >
> > > [1] Lucic, Ana, et al. "Cf-gnnexplainer: Counterfactual explanations for graph neural networks." International Conference on Artificial Intelligence and Statistics. PMLR, 2022.

---

> > > > ### Comment · Reviewer_Tz8g · 2023-08-21
> > > >
> > > > Considering the authors' reply, I further increased my score to 5!

---

### Official Review · Reviewer_MWPm · 2023-07-06

**Soundness:** 3 good
**Presentation:** 3 good
**Contribution:** 3 good
**Rating:** 6
**Confidence:** 3

**Summary:**

The increasing application of graph neural networks (GNNs) has sparked significant interest in understanding their explainability. In this paper, the authors introduce D4Explainer, a method that provides in-distribution GNN explanations for both counterfactual and model-level explanation scenarios. By integrating generative graph distribution learning into an optimization objective, D4Explainer accomplishes two things: 1) It generates a set of diverse counterfactual graphs that comply with the in-distribution property for a given instance; and 2) It identifies the most discriminative graph patterns that contribute to a specific class prediction, thereby providing model-level explanations.

**Strengths:**

1. The paper is generally well-written, clearly outlining the motivations and intuitions behind the proposed method.
2. The concept of employing a diffusion process for graph explanation is both timely and effective.
3. The superior performance of the proposed method is well substantiated by diverse experiments in different settings.

**Weaknesses:**

1. Certain technical details are glossed over. While the page limit is a concern, it's crucial that the key concepts are well explained in the main paper to ensure it is self-contained. For instance, even though Equation (2) is one of the key components of the proposed method, its derivation process and the meanings of the terms within the equation are not clearly presented, which could potentially pose challenges for those who are not experts in graph diffusion.
2. The authors claim that D4Explainer is a unified framework for counterfactual explanation and model-level explanation, but the resulting approaches seem distinct and separate. I would expect more justification on why D4Explainer can be considered a unified framework, and if it could be further extended to tackle other related tasks in graph explanation.
3. Considering the high complexity of the proposed method, an ablation study (or a hyperparameter sensitivity study) would be beneficial for the counterfactual explanation task to provide a clearer understanding of the success of the proposed method. In addition, how did the authors tune hyperparameters (apart from just T and K) for their approach and the competitors?

**Questions:**

1. I’m interested in understanding the technical contributions of the proposed method, especially the generation of counterfactual graphs via a diffusion process. Which aspects of Figure 2a derive from existing works and which are novel proposals in this work?
2. Why is the GIN used solely for the Mutag dataset, whereas the GCN is utilized for all other tasks?

**Limitations:**

The authors have outlined a future research direction.

---

> ### Author Rebuttal · Authors · 2023-08-10
>
> We thank the reviewer for constructive feedback and positive comments on paper writing, architecture efficiency, and extensive experiments. We address the potential concerns as follows.
> >Certain technical details are glossed over. Some equations may not be elaborated in detail.
>
> Thanks for bringing this up. We agree we didn’t elaborate on each equation in detail due to the space limitation, leaving them in the appendix. In our revised version, we will add more descriptions and detailed derivations for important equations to ensure each key concept is well explained and elaborated.
> For your reference, we will add the following content to explain Eq(2). In Eq(2), the objective of $L_{dist}$ is to guarantee the proximity of generated counterfactual explanations to the original graph. Let $A_0$ and $A_t$ denote the corresponding adjacency matrix of graph $G_0$ and $G_t$. $L_{dist}$ is equivalent to the expected cross-entropy loss between $A_0$ and $A_t$, where $G_t$ follows the distribution of $q(G_t|G_0)$.
> >Why is D4Explainer a unified framework for graph explanation tasks? Is there any further extension to other related tasks?
>
> The unification of D4Explainer lies in the same diffusion process and denoising model for different explanation scenarios. The differences between D4Explainer on counterfactual and model-level explanation tasks are (1) loss function and (2) reverse sampling process. Specifically, the loss function for the model-level explanation task does not contain $L_{cf}$, which is designed to ensure the counterfactual property. Moreover, the reverse sampling process in the model-level explanation tasks utilizes multiple-step sampling to increase the confidence score of generated model-level explanations.
>
> We believe the flexibility in the loss function and reverse sampling process does not diminish the overall unification of D4explainer in different explanation tasks. Instead, these flexibilities enable D4Explainer to tackle other related explanation scenarios. For example, D4Explainer is able to generate instance-level factual explanation, where we aim to generate a succinct and sparse explanatory subgraph $G_e\sim p(G_t)$ from a given instance $G_0$. Accordingly, $L_{dist}$ can be replaced with the cross-entropy loss between $G_0$ and $G_0-p(G_t)$, which ensures the generated subgraph is sufficiently succinct. $L_{cf}$ can be modified to accommodate the factual explanation scenario easily.
>
> Apart from flexible loss functions and reverse sampling strategy, D4Explainer uses a fixed diffusion process and denoising network, enabling a wide range of application scenarios with minimal modifications.
> >About our ablation studies and hyperparameter tuning.
>
> We report the search ranges of main hyperparameters in counterfactual explanations as follows. The best hyperparameters have been reported in Appendix E.3. Our ablation studies also show that increasing the number of layers and number of hidden units in the denoising network potentially improves the model performance, while the model performance is not sensitive to dropout rate and alpha.
> | hyperparameter            | search range                              |   |
> |---------------------------|-------------------------------------------|---|
> | number of layers in PPGN  | 3,4,5,6,7,8                               |   |
> | number of hidden features | 32, 64, 128                               |   |
> | alpha                     | 1e-3, 5e-3, 1e-2, 5e-2, 0.1, 0.5, 1, 2, 5 |   |
> | dropout rate              | 0.1, 0.2, 0.3,0.4,0.5                     |   |
> | learning rate             | 1e-4, 5e-4, 1e-3, 5e-3, 1e-2, 5e-2        |   |
>
> >What are the technical contributions of D4Explainer? Which aspects of Figure 2a are novel in this work?
>
> In Figure 2a, we adapt the diffusion process defined in [1,2] to the graph explanation task. We also carefully designed the denoising model to process the noise level indicator which enhances its denoising capability for graph data specifically (refer to Appendix D.1 for more details). Another core innovation in Figure 2a lies in the loss function. We proposed an effective loss function for generating high-quality and in-distribution explanations for GNNs by incorporating graph distribution learning into the optimization objective. We believe these are non-trivial contributions to the graph explainability literature.
>
> [1]DiGress: Discrete Denoising diffusion for graph generation
>
> [2]Diffusion Models for Graphs Benefit From Discrete State Spaces
>
> >Why is the GIN used for Mutag and GCN used for others?
>
> In our implementation, vanilla GCNs do not perform well (<80% accuracy) on the Mutag dataset, Therefore, we use GIN as the target GNN on the Mutag dataset following previous works [1], which achieves 87% in terms of classification accuracy.  We believe a well-performing GNN can also benefit the explanation quality, while it’s worth mentioning that our explainability framework is architecture-agnostic. Thanks again for lighting up this interesting question.
>
> [1]Reinforced Causal Explainer for Graph Neural Networks

---

### Official Review · Reviewer_XFfB · 2023-07-06

**Soundness:** 4 excellent
**Presentation:** 3 good
**Contribution:** 4 excellent
**Rating:** 7
**Confidence:** 4

**Summary:**

A diffusion-based approach for generating instance-level
counterfactual explanations and model-level explanations of GNN
predictions.

**Strengths:**

The approach improves over existing counterfactual-based solutions by
allowing both addition and removal of nodes, thus better retaining
in-distribution of explanations.

An extensive experimental evaluation shows consistent and substantial improvements over existing alternatives.

**Weaknesses:**

The presentation is a bit condensed, so that one has to jump to the
supplementary to get a clear understanding of the novelty of the
proposed solution wrt the works it builds upon.

**Questions:**

In the model-level explanation setting, I wonder whether there is a way to predict the best number of nodes for the generated explanation.

**Limitations:**

Lack of management of node/edge attributes, which the authors do recognize and leave as future work.

Lack of automated selection of number of nodes in model-level explanations (same as XGNN).

---

> ### Author Rebuttal · Authors · 2023-08-10
>
> We thank the reviewer for the positive comment on the proposed model and extensive experiments. We sincerely appreciate the feedback and suggestions you provided, which help us to further improve the quality of this paper. We have addressed each point and potential concern you raised as follows.
> >The presentation is a bit condensed. A comprehensive understanding of the novelty requires referring to the supplementary material.
>
> Thanks for your constructive suggestions. We agree that in our initial version, some differences between the proposed solution and previous works were elaborated in the appendix, due to the space limitations. In our revised version, we ensure each point of our novelty and distinctions from previous works is well explained and mentioned in the main text.
> >Is there a way to predict the best number of nodes for generated model-level explanation?  Lack of automated selection of the number of nodes in model-level explanation.
>
> Thanks for the interesting question. Actually, automatically generating a succinct model-level explanation without a predefined number of nodes is feasible with the current framework. To achieve that, we can put the number of non-isolated nodes into the selection policy. In this way, D4Explainer tries to find the best tradeoff between confidence score and explanation size. However, we tend to believe this manner will diminish the diversity of the generated explanations and easily generate wrong or weak results, which explains why we set the number of nodes as an initialization hyper-parameter, as XGNN[1].
>
> Actually, the definition of the “best” number of nodes in model-level explanations is ambiguous. In many real-world scenarios, the ground truth model-level explanations are not unique. That is, we can hardly know the exact discriminative graph structure/feature that the GNNs learned for prediction [2]. Table 10 in Appendix E.8 shows that model-level explanations with different numbers of nodes can achieve comparably high confidence scores, which indicates that D4Explainer always tries to find the best graph structures and discriminative features for a certain class, given any number of nodes in the desired explanation.
>
> [1]XGNN: Towards Model-Level Explanations of Graph Neural Networks
>
> [2]GNNInterpreter: A Probabilistic Generative Model-Level Explanation for Graph Neural Networks
>
> >Lack of management of node/edge attributes.
>
> Theoretically, the diffusion process and denoising network can indeed be applied to both discrete adjacency matrix and continuous/categorical node/edge attributes. We’ve discussed the discrete diffusion process in the main text and Appendix C.1, which can be applied to process discrete adjacency matrices and categorical features. We also discussed the continuous diffusion process in Appendix C.2, which can be used to manage continuous node/edge features. As mentioned in L132-133, we mainly focus on discrete graph structures and investigate how they affect counterfactual properties and model-level explanation in this work. If time allows, we are more than happy to incorporate empirical results with node/edge attributes in our final version to enrich the analysis.

---

### Decision · Program_Chairs · 2023-09-21

**Decision:**

Accept (poster)

**Comment:**

All reviewers tend towards accepting the paper. The strengths of the paper are improvements over existing counterfactual-based solutions by allowing both the addition and removal of nodes, extensive experiments, and a novel and interesting idea of using diffusion for explaining GNNs.